# An AGO10:miR165/6 module regulates meristem activity and xylem development in the Arabidopsis root

Shirin Mirlohi, Gregory Schott, André Imboden & Olivier Voinnet ✉

## Abstract

The RNA-silencing effector ARGONAUTE10 influences cell fate in plant shoot and floral meristems. ARGONAUTE10 also accumulates in the root apical meristem (RAM), yet its function(s) therein remain elusive. Here, we show that ARGONAUTE10 is expressed in the root cell initials where it controls overall RAM activity and length. ARGONAUTE10 is also expressed in the stele, where post-transcriptional regulation confines it to the root tip's pro-vascular region. There, variations in ARGONAUTE10 levels modulate metaxylem-vs-protoxylem specification. Both ARGONAUTE10 functions entail its selective, high-affinity binding to mobile miR165/166 transcribed in the neighboring endodermis. ARGONAUTE10-bound miR165/166 is degraded, likely via SMALL-RNA-DEGRADING-NUCLEASES1/2, thus reducing miR165/166 ability to silence, via ARGONAUTE1, the transcripts of cell fate-influencing transcription factors. These include PHABULOSA (PHB), which controls meristem activity in the initials and xylem differentiation in the pro-vasculature. During early germination, PHB transcription increases while dynamic, spatially-restricted transcriptional and post-transcriptional mechanisms reduce and confine ARGONAUTE10 accumulation to the provascular cells surrounding the newly-forming xylem axis. Adequate miR165/166 concentrations are thereby channeled along the ARGONAUTE10-deficient yet ARGONAUTE1-proficient axis. Consequently, inversely-correlated miR165/166 and PHB gradients form preferentially along the axis despite ubiquitous PHB transcription and widespread miR165/166 delivery inside the whole vascular cylinder.

**Keywords** AGO1-vs-AGO10 Competition; Gradient; Mobile miR165/166; PHABULOSA (PHB); Root Apical Meristem (RAM)
**Subject Categories** Development; Plant Biology; RNA Biology

See also: N El Arbi et al

## Introduction

Eukaryotic ARGONAUTEs (AGOs) form a highly conserved class of RNA-silencing effector proteins involved in endogenous gene regulation and defense against invading nucleic acids such as viruses, transposons, and transgenes (Dong et al, 2022; Lecellier and Voinnet, 2004; Vaucheret and Voinnet, 2023). Of the ten Arabidopsis AGO paralogs (Vaucheret, 2008), AGO1 regulates gene expression using endogenous microRNAs as its main cargoes (Bologna and Voinnet, 2014; Poulsen et al, 2013). Plant primary miRNA transcripts (pri-miRNAs) are processed by DICER-like-1 (DCL1) into mature 21–24-nt miRNA species. Loaded into AGO1, they guide endonucleolytic cleavage and/or translational repression of miRNA-sequence-complementary transcripts involved in many biological processes including developmental pattering and stress adaptation (Bologna and Voinnet, 2014; Voinnet, 2009). AGO1's closest paralog, AGO10, also known as ZWILLE (ZLL)/PINHEAD (PHD), was first identified for its role in shoot apical meristem (SAM) maintenance due to the fact that strong *ago10* mutants undergo terminal SAM differentiation, resulting in a pinoid-like structure precluding further growth in the Ler background (Lynn et al, 1999; Mallory et al, 2009; Moussian et al, 1998; Tucker et al, 2008). Unlike that of the near-ubiquitous AGO1, AGO10 accumulation in the SAM is restricted to the vasculature, beneath, and on the adaxial side of lateral organ primordia, where it regulates their polarity (Aichinger et al, 2012; Liu et al, 2009; Tucker et al, 2008; Zhu et al, 2011). Both SAM maintenance and organ polarity functions of AGO10 entail fine-tuning of AGO1:-miR165/166-mediated silencing of *HD-ZIP III* transcription factor mRNAs (Jung and Park, 2007; Liu et al, 2009; Zhou et al, 2007; Zhu et al, 2011). In the SAM, AGO10-bound miR165/166 are not only prevented from loading into AGO1, but also undergo degradation via AGO10-coupled SMALL-RNA-DEGRADING-NUCLEASES1/2 (SDN1/2) activities (Ramachandran and Chen, 2008; Yu et al, 2017). AGO10 also accumulates in the floral meristem (FM), where, likewise, it intercepts an AGO1:miR165/166:*HD ZIP III* module and quenches its biological output, including the regulation of the floral stem cell specifier APETALA2 (AP2; (Ji et al, 2011). However, contrary to its role in the SAM where it promotes meristem activity/maintenance, AGO10 stimulates stem cell termination in the FM, by indirectly antagonizing AP2 function (Ji et al, 2011).

---

Department of Biology, Swiss Federal Institute of Technology (ETH-Zürich), Universitätsstrasse 2, 8092 Zürich, Switzerland. ✉E-mail: voinneto@ethz.ch

Despite these contrasting biological outputs, both SAM- and FM-functions of AGO10 entail a uniquely high affinity and specificity for miR165/166 (Ji et al, 2011; Yu et al, 2017; Zhu et al, 2011), the bases of which have been recently clarified by reconstructing the selective affinity of recombinant Arabidopsis AGO10 for synthetic miR166a in vitro (Xiao and MacRae, 2022). The process was found to require a complex combination of factors most likely relevant in vivo, which, as discussed (Xiao and MacRae, 2022), may be uniquely found in some (e.g. SAM, FM) unlike other tissues. Added to miR166a-intrinsic sequence/structure determinants, extrinsic factors include optimal physiological concentrations of sulfate and phosphate as well as a tight balance of co-chaperones HSP70 and HSP90 exerting distinct yet coordinated effects on AGO10 loading and selectivity (Xiao and MacRae, 2022). Further complicating this picture, miR398c, which, by sequence and structure, is unrelated to miR165/166, also undergoes selective AGO10-mediated sequestration in the ovule's chalaza (Cai et al, 2021), suggesting altogether, tissue- or organ-dependent contexts to miRNA selectivity by AGO10.

One organ in which putative high-affinity interactions between AGO10 and miR165/166 (or other miRNAs) might occur, is the Arabidopsis root. There, AGO10 was recognized as a core identity-marker of the stele (Iyer-Pascuzzi et al, 2011) in which it indeed accumulates and its levels fluctuate in an abiotic stress-responsive manner (Bloch et al, 2019; Iki et al, 2018; Palovaara et al, 2017). This suggests important, albeit as yet experimentally untested functions for AGO10 in this, and possibly other root layers. The stele is composed of an outer pericycle ring encircling a vascular cylinder occupied by perpendicularly positioned phloem and xylem vascular cells separated by a mass of procambium cells. The endodermis, cortex and epidermis surround the stele in concentric layers. All layers are constantly replenished by dividing stem cells called initials, which derive from, and are adjacent to, the quiescent center (QC) located at the very tip of the root (Dolan et al, 1993). The stele-proximal endodermis and stele-distal cortex share a common "ground tissue" initial, as do the outer epidermis and lateral root cap. Directly above and below the QC are the stele and columella initials, respectively (Lee et al, 2013; Rahni and Birnbaum, 2019). While AGO10 functions in the root remain largely elusive/speculative, the AGO1:miR165/166:*HD ZIP III* module has been implicated in regulating the root apical meristem (RAM) activity/length (Dello Ioio et al, 2012; Singh et al, 2014) and in specifying meta-*vs* proto-xylem (MX-*vs*-PX) cells along the stele's xylem axis (Carlsbecker et al, 2010; Miyashima et al, 2011). The latter process involves the activation of *MIR165/166* transcription by SHORTROOT (SHR) and SCARECROW (SCR) exclusively in the endodermis, and the ensuing movement of miR165/166 to the neighboring stele (Carlsbecker et al, 2010; Miyashima et al, 2011) and cortex/epidermis (Brosnan et al, 2019). Experimental evidence suggests that either mature miRNAs or pri-miRNAs can move between cells (reviewed in Voinnet; 2022) and the form of miR165/166 movement remains unclear in this regard. One expected consequence of movement of mature miR165/166, as opposed to pri-miR165/166, would be their progressive "consumption" upon their loading into relevant AGOs. AGO1 would consume miR165/166 via PTGS in a manner likely proportional with the amount of *HDZIP III* target mRNAs found in traversed cells, as previously anticipated (Carlsbecker et al, 2010). This target-directed miRNA degradation (TDMD; Shi et al, 2020) would be

likely seconded, inside the stele, by enhanced turnover of the AGO10-bound miR165/166 fraction via SDN1/2 activities, assuming that AGO10 plays in the RAM the same miR165/166-quenching function exerted in the SAM and FM.

Here, we have compared AGO10's transcription- and protein accumulation-patterns to understand how the protein remains highly confined within discrete regions of the RAM. By genetically reducing/enhancing AGO10 levels within its cognate root expression domain, we have explored if, as anticipated, AGO10 regulates the RAM length/activity, and if this is achieved by impeding (as in the FM) or promoting (as in the SAM) stemness. We have also explored, in parallel, if AGO10 indeed modulates MX-*vs*-PX development in the stele. We have further queried if regulations of RAM length/activity and MX-*vs*-PX development entail AGO10's selective and competitive affinity for mobile miR165/166 over AGO1, as well as miR165/166 degradation. We have explored which RAM's cell type(s) and which subcellular compartments thereof might possibly underly the proposed AGO10-*vs*-AGO1 competition, particularly in relation with the as-yet-undetermined mobile form(s) of miR165/166. We have asked, finally, if and how new information on the spatio-temporal distribution of AGO10 in the RAM might help refine a model for *HD-ZIPIII* gradient formation along the root tip's xylem axis. The results of this study expand our understanding of root vascular development and RAM maintenance, and illustrate how critical spatio-temporal positioning of an AGO protein can shape and refine a mobile miRNA activity gradient across multiple cell types.

# Results and discussion

## Post-transcriptional regulation restricts AGO10's vascular accumulation to the division zone of the root apical meristem

To study AGO10 localization as an indicator of its possible functions in roots, we used Arabidopsis lines (Col-0; *ago10-1*-null background (Takeda et al, 2008)) expressing the *pAGO10::GF-P:AGO10* translational reporter (Iki et al, 2018; Jullien et al, 2022), abbreviated *pA10::G:A10(ago10-1)* (Figs. 1A,B and EV1A). Confocal microscopy conducted six days-after-germination (DAG) of independent T2 lines consistently revealed a prominent signal in the root tip's stele (Fig. 1A), although its intensity varied between lines, likely reflecting transgene copy number and/or genomic context. From their, respectively, high, moderate, and low GFP signals, three T2s were selected to establish stable T3 lines (#1–3). Western blot analyses conducted with an anti-AGO10 antibody (Grentzinger et al, 2020) revealed that AGO10 accumulates ~5 times less in whole roots than in whole inflorescences (Fig. 1B). In roots, line#1 displayed the highest GFP:AGO10 levels (Fig. 1B, green arrow) on par with those of endogenous (endo)AGO10 in WT non-transgenic inflorescences and, hence, ~5 times higher than those of endoAGO10 in WT non-transgenic roots (Fig. 1B, blue arrows). Presumably due to these high expression levels, a GFP:AGO10 degradation fragment accumulated in line #3 (Fig. 1B, green asterisk), which was also detected in inflorescences under longer exposure (Fig. EV1B). In whole roots, line#2 and #3 accumulated, respectively, moderate and low GFP:AGO10 levels compared to line#1; those in line#2 were comparable with, and

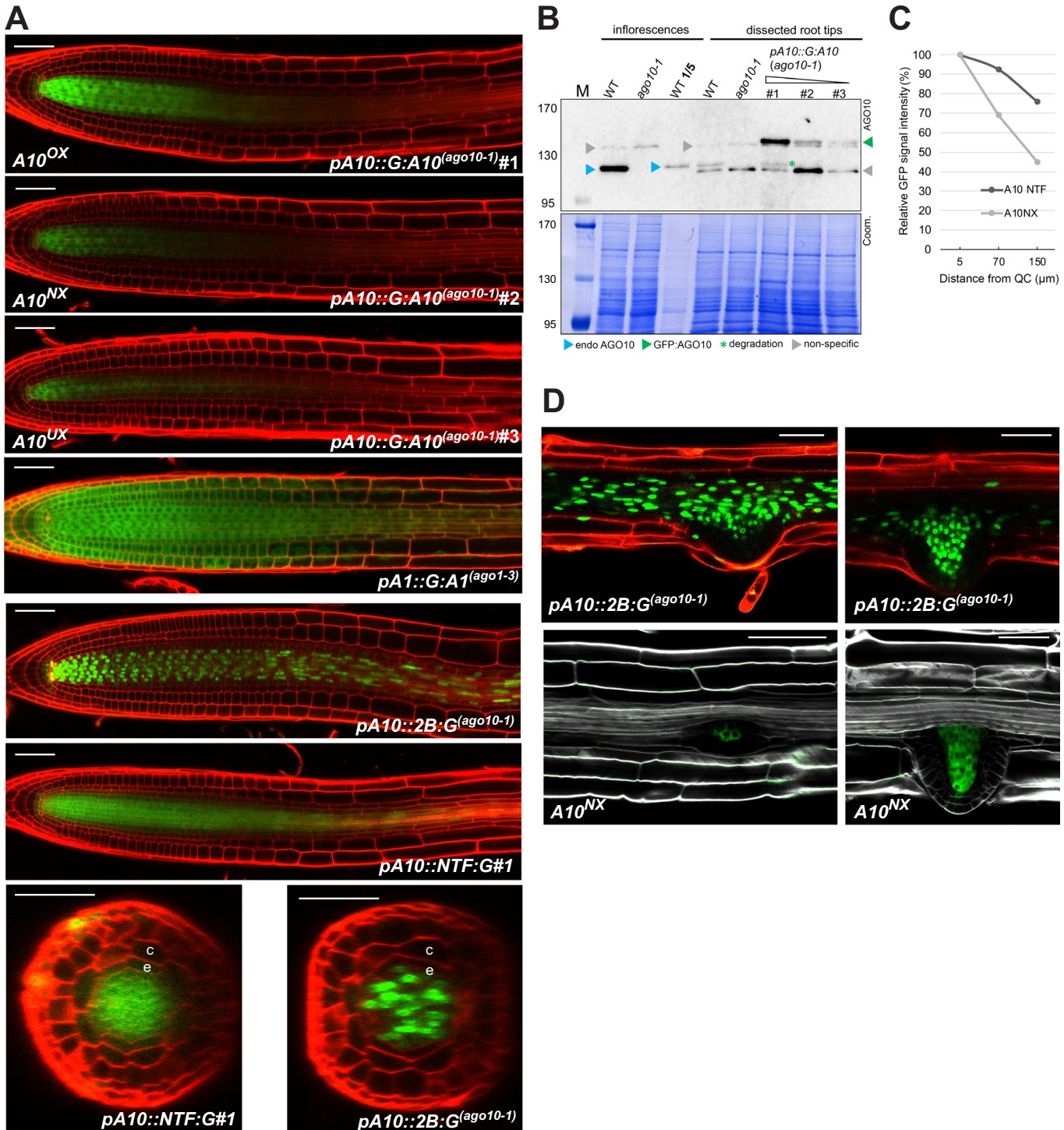

those of line#3 below endoAGO10 levels (Figs. 1B and EV1B). All analyses described in this study are consistent with the notion that line#2 (referred to as $A10^{NX}$) reports the cognate AGO10 root expression domain/level, whereas line#1 and #3 are AGO10 over-($A10^{OX}$) and under-($A10^{UX}$) expressors within this cognate expression domain, respectively. In all three lines, the stele-prominent GFP:AGO10 signal was highest in the QC-proximal division zone and gradually faded toward the elongation zone where it eventually

reached below-detection levels as in the entire differentiation zone and beyond (Fig. 1A). By contrast, $pAGO1::GFP:AGO1^{(ago1-3)}$ (abbreviated $pA1::G:A1^{(ago1-3)}$) was detected in all layers, spanning, as reported (Bologna et al, 2018), the division, elongation, and differentiation zones (Figs. 1A and EV1A). Therefore, AGO10 accumulates predominantly in the stele of the root apical meristem (RAM), inside which its expression overlaps with that of its closest and ubiquitously expressed paralog, AGO1.

◄ **Figure 1. AGO10 accumulates in the root tip's vasculature.**

(A) GFP signals yielded in root tips by the indicated reporters in the indicated genetic backgrounds, in longitudinal views. Cell walls were stained with PI, yielding a red signal. A10$^{NX}$ root picture re-used in Fig. EV1A and pA10::2B:G $^{(ago10-1)}$ re-used in Fig. EV1C. (B) Western analyses of GFP:AGO10 (green arrow) and endoAGO10 (blue arrow) accumulation in inflorescences (where the anti-AGO10 antibody yields almost no unspecific or background signals) or roots (yielding more non-specific signals; gray arrows) of WT, ago10-1 and pA10::G:A10$^{(ago10-1)}$#1,2,3 plants. The 'WT inflorescences' sample in track 4 was diluted 5 times to enable better comparison with the endoAGO10 levels in roots. Green asterisk: GFP:AGO10 degradation product also accumulating in inflorescences (see also Fig. EV1B). Coom: Coomassie blue staining provides a total protein loading control. M: protein ladder with molecular weight in kDa. The experiment was repeated twice with similar results. (C) GFP signal intensity from lines A10$^{NX}$ and pA10::NTF:G#1 measured along the root longitudinal axis at various distances from the QC. Each dot represents independent measurements involving 10 plants for each background. See Dataset EV1 for the underlying quantitation. (D) Lateral pA10::2B:G$^{(ago10-1)}$ roots (top panels) or A10$^{NX}$ (bottom panels) roots, reporting, respectively, the (pro)vascular AGO10 transcription and GFP:AGO10 accumulation. All images are representative of at least $n = 20$ independent confocal observations; all scale bars: 50 μm. c: cortex; e: endodermis. (A) Roots have been inspected at 6 DAG. (D) Early (left) and more advanced (right) stages of lateral root formation in mature roots of the respective backgrounds are shown. Source data are available online for this figure.

To explore the *AGO10* gene expression pattern, we engineered the *pAGO10::H2B:GFP* transcriptional reporter into *ago10-1* Arabidopsis, creating *pAGO10::H2B:GFP(ago10-1)*, abbreviated *pA10::2B:G(ago10-1)*. The nuclear H2B:GFP signal facilitates studies of low-expressed genes and prevents GFP intercellular movement. 20-out-of-20 GFP-positive T2s displayed the same 6-DAG signal in primary root tips (Figs. 1A and EV1C). The signal was most prominent in the stele along the radial axis. Along the longitudinal axis, the signal from *pA10::2B:G(ago10-1)* in the stele extended well above the division zone into the elongation and differentiation zones, and was still detected in the mature root. By contrast, the signal from the translational reporter *pA10::G:A10(ago10-1)* was below detection in all these regions (Fig. 1A, EV1C). H2B:GFP's half-life unlikely underlies these signals' discrepancy because the same pattern was observed with *pAGO10* driving expression of a GFP-tagged nuclear-targeting fusion (NTF) protein associated with the nuclear envelope (Palovaara et al, 2017) (*pA10::NTF:G*#1; #2; Figs. 1A and EV1C). GFP signal quantification suggested that both *pA10::NTF:G*#1 and *pA10::G:A10 (ago10-1)* form a gradient from the QC-proximal-to-QC-distal region of the RAM (Fig. 1C; Dataset EV1). However, the latter gradient is steeper, suggesting that GFP:AGO10 undergoes post-transcriptional negative regulation. A time-course analysis of GFP:AGO10 accumulation during early germination (Fig. 6A) supports this notion. In addition, transiently-expressed GFP:AGO10 and GFP:AGO1 accumulate to similar levels in *N.benthamiana* leaves (Fig. EV1D). This contrasts with the distinct accumulation patterns of each fusion protein within the stele in the Arabidopsis RAM (Fig. 1A). Thus, a stele-intrinsic biological process, as opposed to selective destabilization caused by fusing GFP, likely underpins the steep gradient observed therein with GFP:AGO10, but not with GFP:AGO1. The henceforth suggested negative regulation of AGO10 is probably reversible, as hinted in the mature root where *AGO10* transcription appears constitutive in both the main stele and that of developing lateral roots (Fig. 1D). AGO10 protein accumulation, by contrast, is circumscribed to the meristematic vasculature of developing lateral roots (Fig. 1D), which are initiated from the main stele (Dolan et al, 1993).

## AGO10 controls the RAM activity/length likely by modulating the extent of *PHB* silencing via the non-cell-autonomous action of miR165/166

A first hint regarding AGO10 function(s) in the RAM came from observations that *ago10-1* root tips have an increased meristem length compared with non-transgenic WT- or A10$^{NX}$-root tips

(Fig. 2A,B). Conversely, *A10$^{OX}$* plants display shorter meristems evoking the *shortroot* (*shr*) mutant phenotype (Helariutta et al, 2000) (Fig. 2A,B). The cell number on the longitudinal plan in individual ground tissue files (cortex and endodermis) was, respectively, decreased in *A10$^{OX}$*- and increased in *ago10-1* roots, as compared with their non-transgenic WT- or *A10$^{NX}$*-counterparts (Fig. 2C,D; Dataset EV2). Respectively, more-*vs*-less cells were also observed in both the QC-proximal and QC-distal zones of the stele in *ago10-1-vs-A10$^{OX}$*-root tips' radial plan (Fig. 2E,F; Dataset EV2). In principle, this increased cell number could result from either enhanced cell division or delayed differentiation. Of these two possibilities, we favor the former because (El Arbi et al 2024) show in their accompanying study that both anticlinal and periclinal divisions are increased in the *ago10* mutant's stele.

miR165a/b and its 1-nt sequence variant, miR166a/b, are the most abundant miR165/166 paralogs in the Arabidopsis root tip (Brosnan et al, 2019). Strikingly, the opposite RAM length/activity-defects observed here in *ago10-1-vs-A10$^{OX}$* phenocopy those of Arabidopsis roots previously engineered to either overexpress miR166a, or to neutralize its activity (Singh et al, 2014). miR166a overexpression increased RAM length/activity as seen here in *ago10-1*; conversely, neutralizing miR166a activity via target-mimicry reduced RAM length/activity, as seen here in *A10$^{OX}$* (Singh et al, 2014). miR166a neutralization also caused an expansion of the accumulation domain of *PHABULOSA (PHB)* (Singh et al, 2014), which promotes differentiation in the RAM (Dello Ioio et al, 2012) and is negatively regulated by miR165/166 among several sequence-related *HD-ZIP III* transcription factors. SCARECROW (SCR)-dependent and endodermis-restricted transcription of *MIR166a/b* and *MIR165a/b* is elicited by SHR, which moves from the stele (Carlsbecker et al, 2010). Strikingly, *shr-2* mutant roots display reduced RAM length/activity (Helariutta et al, 2000), like those of *A10$^{OX}$* (Fig. 2A). Assuming that, as seen in the SAM (Zhou et al, 2015; Zhu et al, 2011), AGO10 selectively quenches AGO1:miR165/166-mediated *HD-ZIP III* silencing in the RAM, the above observations can be rationalized in the model depicted in Fig. 2G. In this model, loss-of-AGO10 in the RAM would enhance AGO1-mediated *PHB* silencing and, hence, increase RAM length/activity. Conversely, gain-of-AGO10 would reduce *PHB* silencing, decreasing RAM length/activity (Fig. 2G).

In *A10$^{OX}$* and *ago10-1*, the cell numbers are altered in layers where GFP:AGO10 is below detection, except in the stele (Fig. 2A–D). This prompted us to explore AGO10 expression in initials, the QC-derived stem cells that continuously replenish each layer (Dolan et al, 1993; Lee et al, 2013; Rahni and Birnbaum, 2019). A signal from a *pA10::2B:G(ago10-1)* transcriptional reporter

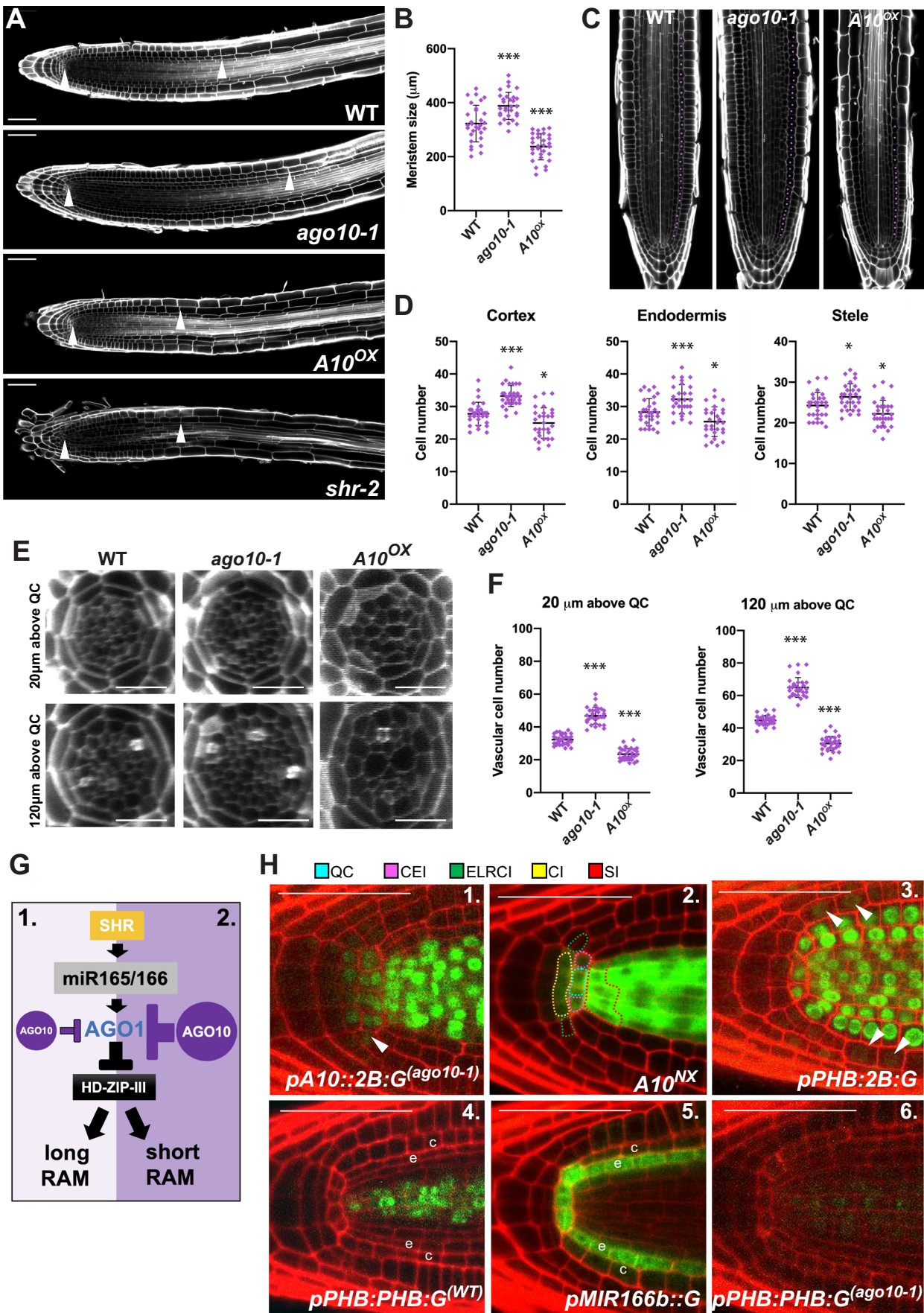

**Figure 2. AGO10 modulates the RAM length/activity.**

(A, B) Meristem length in the indicated genotypes measured by the distance between the QC (left arrows) and first elongating cortex cells (right arrows). (C, D) Cell number quantification in the indicated genotypes' cell layers. The middle bar in (C) (300 μm) confirms counting over equal areas. (E, F) Images of the stele in the indicated genotypes measured at 20 μm or 120 μm from the QC. (G) Model for AGO10-mediated modulation of RAM length/activity influenced by the extent of *HD-ZIP III* silencing. See text for details. (H) QC-proximal views of the signals yielded by the indicated reporters in the specified genetic backgrounds. The arrow in panel 1 indicates a signal in the ELRCI. In panel 2, the initials of each root cell layer are color-outlined with dotted lines according to the legend provided. The arrows in panel 3 indicate signals in the ELRC and epidermis cells derived from it. e: endodermis, c: cortex. SI: stele initials; CEI: cortex/endodermis initial; QC: quiescent center; ELRCI: epidermis/lateral root cap initial; CI: columella initial. Scale bars in (A), (E) and (H): 50 μm. Data information: For panels (B), (D), (F): black bars: mean. Error bars: standard deviations. (student t-test *p < 0.05, **p < 0.005, ***p < 0.0005). cell numbers calculated in individual plants (n = 30). See also Dataset EV2 for specific cell numbers' quantitation in individual plants (A–F) Roots have been inspected at 6 DAG. (H) All images are representative of at least n = 20 independent confocal observations. Source data are available online for this figure.

was indeed detected in the QC and the initials of the columella (CIs), cortex-endodermis (CEIs), and stele (SIs) (Fig. 2H, #1). *pA10::2B:G$^{(ago10-1)}$* also yielded a signal in the epidermis-lateral root cap initials (ELRCIs) albeit at lower levels (Fig. 2H #1, arrow). The signal was below detection, however, in cells directly derived from all initials (e.g., transit amplifying cells), except those derived from the SI, in which it was strong (Fig. 2H #1). *A10$^{Nx}$* yielded a similar overall pattern, although the signal was below detection in the ELRCIs consistent with lower *pA10::2B:G$^{(ago10-1)}$* expression therein (Fig. 2H #2). To explore *HD-ZIP III* accumulation in the meristem and initials, we engineered transgenic Arabidopsis (WT background) expressing either the *pPHB::H2B:GFP* transcriptional reporter (*pPHB::2B:G*) or a previously-described *pPHB::PHB:GFP* (*pPHB::PHB:G*) translational reporter (Miyashima et al, 2011).

*pPHB::2B:G* yielded strong transcriptional signals in the QC, CEIs, SIs and cells derived from them; it was fainter, though detectable, in the ELRCIs and early epidermis cells derived thereof (Fig. 2H #3; arrows). By contrast, in four independent GFP-positive T2 lines, the *pPHB::PHB:G* signal was only detected in the SIs and the stele, indicating post-transcriptional negative regulation (Figs. 2H #4 and EV5A,B). The signal seemed to gradually decrease from the stele's inner part toward the endodermis, consistent with earlier observations of *pPHB::PHB:G* in root tips (Carlsbecker et al, 2010; Miyashima et al, 2011). This pattern had been ascribed to miR165/166 generating an inverse gradient of *PHB* and related *HD-ZIP IIIs* in the stele as it moves away and gets progressively diluted from the endodermis, where it is exclusively transcribed (Carlsbecker et al, 2010; Miyashima et al, 2011). We found that the QC also expresses miR165/166 and that AGO1, the effector of miR165/166-mediated silencing, accumulates ubiquitously throughout the RAM and lateral root meristems (Figs. 2H #5 and EV2A,B). Therefore, a role for mobile miR165/166 can now be extended to the initials –all directly adjacent to miR165/166-producing cells– to rationalize the lack of *pPHB::PHB:G* signal in the QC, CEIs, and ELRCIs despite robust *PHB* transcription in these cells. AGO10 accumulating at comparably much higher levels in the SI (Fig. 2H #2) likely explains the detectable levels of *pPHB::PHB:G* therein (Fig. 2H #4) due to the proposed quenching of AGO1:miR165/166-mediated silencing by AGO10. This also likely explains the detectable *pPHB::PHB:G* accumulation in the AGO10-rich stele (Fig. 2H #4). Consistent with these interpretations, the GFP signal, while remaining invisible in the QC, CEIs, and ELRCIs, was strongly reduced in the SIs and stele of four independent *pPHB::PHB:G$^{(ago10-1)}$* lines (Figs. 2H #6 and EV5A,B). We could not test the expected inverse effect in *A10$^{OX}$* (i.e. a gain of *pPHB::PHB:G* in all initials) because the *A10$^{OX}$* background already

yields a GFP signal. Nonetheless, we note that a miR165/166-resistant PHB:GFP allele—which should genetically approximate *A10$^{OX}$*—accumulates in all root tip layers and in the ELRCIs, CEIs, and SIs (Miyashima et al, 2011). Collectively, these and previous observations support the model in Fig. 2G, whereby the influence of *HD-ZIP III* levels on RAM activity/length is modulated in an AGO10-dependent manner likely in miR166/5-receiving initials. However, whether this effect is mostly exerted via cell division control—as shown in the stele by El Arbi et al (2024)—remains to be ascertained in the other initials/layers.

## AGO10's high affinity and selectivity for miR165/166 in the RAM is consistent with its proposed role in meristem activity/length control

As a key underpinning of Fig. 2G's model, AGO10 should display selective and competitive affinity for miR165/166 over AGO1 in the RAM. As explained in the introduction, complex and likely tissue/organ-specific factors underly both AGO10 properties (Xiao and MacRae, 2022). Consequently, they cannot be merely inferred from studies in the SAM or FM (Ji et al, 2011; Liu et al, 2009; Ramachandran and Chen, 2008; Zhou et al, 2015; Zhu et al, 2011). To investigate AGO10's selective and competitive binding to miR165/166 in the RAM, we attempted to immunoprecipitate endoAGO10 using our anti-AGO10 antibody (Grentzinger et al, 2020) (Fig. 1B), albeit unsuccessfully. We resorted to generate GFP-based immunoprecipitates (IPs) using *A10$^{OX}$* root tips where GFP:AGO10 accumulates to higher levels, yet still within its cognate expression domain (Figs. 1A and EV1A). In parallel, we isolated root tips from *pA1::G:A1$^{(ago1-3)}$* and from non-transgenic WT plants. Similar to previous analyses conducted in the SAM (Zhu et al, 2011), the *pA1::G:A1$^{(ago1-3)}$* input material was diluted five folds to compensate for the broad-*vs*-narrow expression domain of GFP:AGO1-*vs*-GFP:AGO10 (Fig. 1A). This granted successful isolation of GFP:AGO10 IPs whose signals were at least 20 times lower than those from GFP:AGO1 IPs. Despite this difference, comparable miR165/166 levels were detected in both IPs (Fig. 3A), suggesting that AGO10 displays substantially higher affinity for these cargoes than AGO1. In the SAM, the AGO10-bound pool of miR165/166 undergoes degradation by SDN1/2 (Ramachandran and Chen, 2008; Yu et al, 2017). Hence, AGO10 competitive binding to miR165/166 is diagnosed by AGO1 IPs containing more miR165/166 in *ago10* mutant than in WT apices (Zhu et al, 2011). miR165/166 levels were likewise higher in AGO1 IPs isolated from *ago10-1* compared to WT-dissected root tips, with comparable AGO1 levels detected in both conditions (Fig. 3B). This result

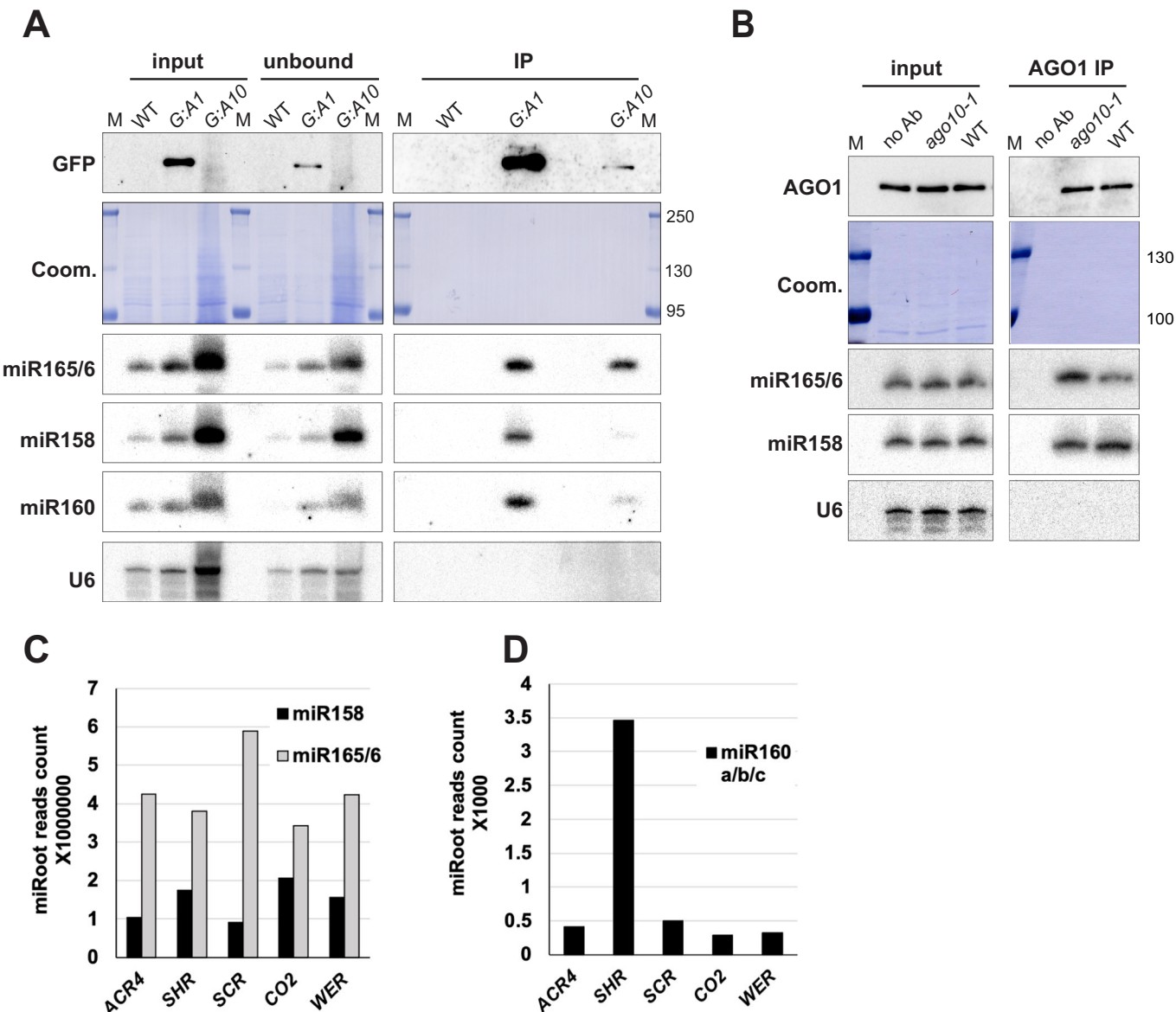

**Figure 3. AGO10 displays high affinity and specificity for miR165/166 in the RAM.**

(A) GFP-based immunoprecipitation (IP) of WT, GFP:AGO1 (*A1*), and GFP:AGO10 (*A10*) in dissected whole root tips at 4 DAG, with the *A1* input material diluted five times compared to *A10*'s. Coom: Coomassie blue staining provides a total protein loading control. M: protein ladder with molecular weight in kDa. The indicated miRNAs were detected by northern blot via hybridization of radiolabeled sequence-complementary oligonucleotides. Hybridization to the U6 snRNA provides a total small RNA loading control. The experiment was repeated twice with similar results. (B) endoAGO1 immunoprecipitation using an anti-AGO1 antibody conducted at 4 DAG in root tips dissected from either WT or *ago10-1* plants. The experiment was repeated twice with similar results. (C) miRoot-based AGO1-loading quantification for the entire miR165/166 family *vs* miR158a/b in the *ACR4, SHR, SCR, CO2,* and *WER* root tip expression domains. (D) Same as (C) for miR160a/b/c. Source data are available online for this figure.

supports the notion that AGO10 competes with AGO1 for miR165/166 binding in the RAM. It further suggests that AGO10 displays higher affinity for miR165/166 than AGO1 in the RAM, given AGO10's much narrower accumulation pattern in root tips (Fig. 1A), contrasting, furthermore, with the pan-layer and near-equal distribution of miR165/166 therein (Fig. 3C).

To explore if, as reported in the SAM (Zhu et al, 2011), AGO10 competition with AGO1 is selective for miR165/166 in the RAM, we analyzed miR158, of which paralog miR158b is the root tip's

most abundant, pan-layer miRNA (Brosnan et al, 2019). Querying AGO1-bound miRNA repertoires within each root layer using miRoot (Brosnan et al, 2019) revealed that miR158 loading into AGO1 within the *SHR* (stele) domain is merely ~2.2 times lower than that of the entire miR165/166 family (Fig. 3C). While miR158 was readily detected in GFP:AGO1 IPs, it barely accumulated in GFP:AGO10 IPs, as did miR160, a much less abundant yet highly stele-enriched AGO1-bound miRNA (Brosnan et al, 2019) (Fig. 3A,D). In addition, the levels of miR158 remained unchanged

in AGO1 IPs isolated from ago10-1 compared to WT root tips, contrasting with the increased levels of miR165/166 (Fig. 3B). Altogether, these results are consistent with GFP:AGO10 displaying high, competitive affinity and selectivity for miR165/166 in the RAM. They support a model whereby an AGO10-vs-AGO1 competition for miR165/166 modulates the extent of *PHB* silencing including in the root layers' initials (Fig. 2G). Figures 1–3 therefore identify AGO10 as a global, *hitherto* unknown RAM length/activity regulator and suggest a novel non-cell-autonomous function for miR165/166 beyond that already recognized in xylem development (Carlsbecker et al, 2010; Miyashima et al, 2011), studied further below.

## AGO10 is required for proper xylem development in the root

Having uncovered a role for AGO10 in the global control of RAM length/activity, we investigated its potential specific function(s) in the stele. There, each pole of the developing xylem axis displays first one outer protoxylem (PX) and later, one larger inner metaxylem (MX) file; a third MX file differentiates centrally further in development (Fig. 4A). In the division zone, a gradient of related HD-ZIP III transcription factors dose-dependently influences PX-*vs*-MX specification: xylem precursor cells with the highest HD-ZIP III levels form MX; those with the lower form PX (De Rybel et al, 2016; Kondo et al, 2014; Ramachandran et al, 2017) (Fig. 4A). As already discussed, the HD-ZIP III gradient is likely established by movement and progressive dilution, from the outer-to-inner stele, of endodermis-derived miR165/166 (Carlsbecker et al, 2010; Miyashima et al, 2011) (Fig. 4A,B case 1). In the RAM, AGO10 displays high affinity for miR165/166 and promotes its degradation (Fig. 3). Furthermore, AGO10 accumulates prominently in the stele of the division zone and its absence/presence therein influences PHB:GFP accumulation (Fig. 2H #4 compared to #6). Thus, AGO10 emerged as a potential additional player in PX-*vs*-MX development, by possibly modulating this process via miR165/166 quenching (Fig. 4B; cases 2a,b). According to this model, ablating/reducing AGO10 expression should enhance *HD-ZIP III* degradation via miR165/166 and thereby promote PX, as opposed to MX formation (Fig. 4B; case 2a); increasing AGO10 expression should yield opposite effects (case 2b).

Basic Fuchsin specifically stains the lignified secondary cell walls of differentiated xylems. In WT root tips, the differentiated PX typically yields a spiral or annular cell wall signal and is visible earlier, i.e. QC-proximally, than the more QC-distal differentiating MX, which yields a perforated cell wall signal (Fig. 4C). To test Fig. 4B's model, root tips of non-transgenic WT, $A10^{NX}$, ago10-1, $A10^{OX}$, or $A10^{UX}$ lines were Fuchsin-stained and xylem specification was scored within an early differentiation zone (QC-proximal) and later within a QC-distal zone, after full differentiation (Fig. 4D, arrows). As a reference for compromised xylem differentiation, we used shr-2 (Carlsbecker et al, 2010) in which *MIR165/166* transcription is substantially reduced, yielding an MX-only phenotype upon Fuchsin staining (Fig. EV3A). In both QC-proximal and -distal zones, one outer PX and one inner MX were observed at each xylem pole (2 PX; 2 MX) in nearly 100% of inspected roots of non-transgenic WT and $A10^{NX}$ plants, as expected (Figs. 4C and EV3B). Fuchsin staining in ago10-1 was more QC-distal than in WT roots (Fig. EV3C), consistent with

ago10-1's increased RAM length/activity (Fig. 2A–C) also expected to delay MX differentiation. In the QC-proximal zone, most ago10-1 roots displayed two PX at each pole (4 PX in total instead of 2 PX in WT) and indeed either zero or one inner MX cells at each pole; two cognate MX cells were less frequently observed (Fig. 4E). In the QC-distal zone, the 4 PX were accompanied by either 2 or 3 MX, with the latter anomaly likely reflecting the increased cell number in the ago10-1's stele (Fig. 2E,F). Xylem defects in QC-proximal and -distal zones of $A10^{UX}$ roots resembled those of ago10-1 and the same was true in roots of sgo1, the ago10 allele isolated by El Arbi et al (2024) in the accompanying manuscript (Fig. EV3D,E). Xylem defects in ago10-1, sgo1 and $A10^{UX}$ essentially phenocopied those of a triple *hd zip III* mutant grown in parallel (*phb phv can*; Fig. 4F), consistent with a role for AGO10 in quenching *HD ZIP III* silencing by miR165/166. These results prompted us to explore if sdn1sdn2 roots display related phenotypes. This was expected if the nucleases' activity was coupled to AGO10 function as shown in the SAM (Ramachandran and Chen, 2008; Yu et al, 2017) and indeed suggested in the RAM (Fig. 3B). In the QC-proximal zone, most sdn1sdn2 roots displayed the cognate 2 PX pattern of WT roots but lacked MX cells altogether. Two cognate MX or, less frequently, only one MX were observed in the QC-distal zone (Fig. 4G). This attenuated ago10-1 phenotype and the results of Fig. 3B therefore support the notion that SDN1/2 facilitate AGO10-mediated quenching of miR165/166 activity in the RAM.

Fuchsin staining also indicated that xylem maturation was more QC-proximal in $A10^{OX}$- than in WT roots (Fig. EV3C), consistent with $A10^{OX}$'s reduced RAM length/activity (Fig. 2A–C). In both the QC-proximal and -distal zones, most $A10^{OX}$ roots lacked PX altogether but displayed the cognate 2 MX pattern of WT roots; less frequently, three MX were observed (Fig. 4H). This phenotype strikingly resembled that of shr-2 in which *MIR165/166* transcription is compromised (Carlsbecker et al, 2010; Fig. EV3A) and of dominant miR165/166-resistant *PHB* alleles (Carlsbecker et al, 2010; Miyashima et al, 2011). Therefore, eliminating/reducing *AGO10* levels yields xylem defects consistent with reduced miR165/166 quenching and hence, increased AGO1-directed *HD-ZIP III* silencing approximating the *phb phv cna* backgrounds' phenotype. Conversely, *AGO10* overaccumulation within its cognate expression domain yields opposite xylem defects consistent with enhanced quenching/degradation of miR165/166 and hence, decreased AGO1-directed *HD-ZIP III* silencing. Collectively, these results support the notion that an AGO10-vs-AGO1 competition for mobile miR165/166 regulates xylem development in the stele (Fig. 4A,B).

## Mobile form of miR165/166 and possible AGO10-vs-AGO1 competition for miR165/166 in different subcellular localizations

Arabidopsis miRNAs can move from cell-to-cell in a tissue/cell-contextual manner (Voinnet, 2022), either as fully processed (i.e. mature) AGO-free molecules (Brioudes et al, 2021; Skopelitis et al, 2018) and/or longer pri-miRNAs subsequently matured in recipient cells (Brosnan et al, 2019; Cai et al, 2021). Whether miR165/166 moves at all in the SAM remains unclear (Yu et al, 2017). By contrast, experimental evidence supports endodermis-to-stele movement of a miR165/166-based entity in the RAM, via plasmodesmata (PDs) (Vaten et al, 2011). The as-yet-undetermined

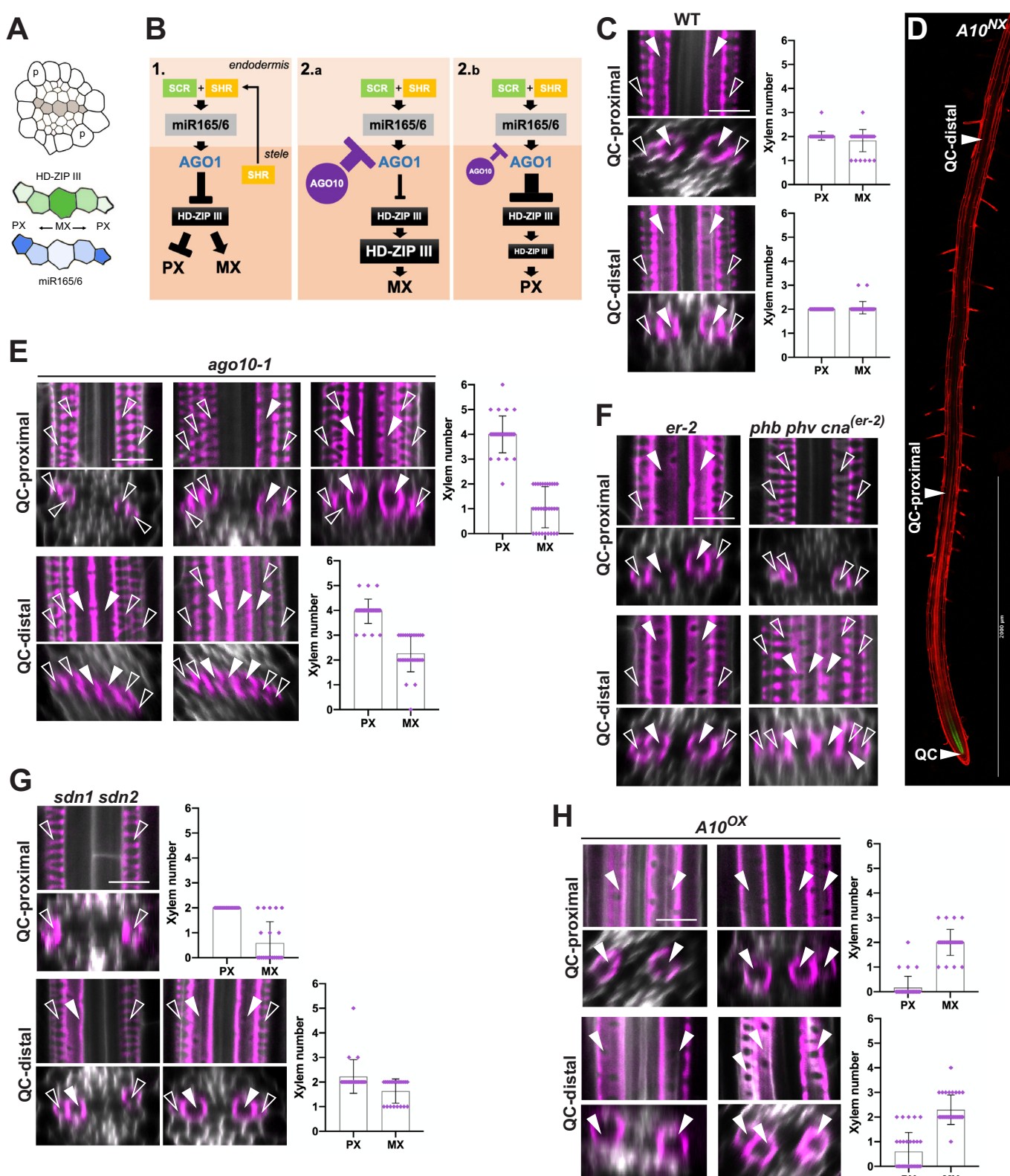

molecular form of this mobile entity matters in the context of the AGO1-*vs*-AGO10 competition for miR165/166 proposed here as a central tenet of our and the accompanying studies (El Arbi et al, 2024). Indeed, gradient formation by miR165/166 would entail

their competitive consumption by both AGOs over traversed cells (Fig. 5A). This would require movement of mature miRNA species, the only known AGO cargoes (Voinnet, 2022). Pri-miR165/166 movement, by contrast, would bypass consumption altogether

**Figure 4.   AGO10 regulates xylem cell differentiation in the root.**

(A) Influence of predicted opposing gradients of mobile miR165/166 (blue)-*vs*-HD-ZIP III (green) concentrations on PX-*vs*-MX differentiation along the root xylem axis (gray in the upper-part stele's schematics). p: pericycle. (B) Model for AGO10-mediated modulation of MX-*vs*-PX differentiation. Panel 1: current model in which SCR and stele-derived SHR activate *MIR165/166* transcription in the endodermis, whereupon miR165/166 movement inside the stele promotes AGO1-dependent *HD-ZIP III* silencing, in turn influencing MX-*vs*-PX formation according to (A). Panels 2.a-b: refined model implicating a stele-based AGO10-*vs*-AGO1 competition for mobile miR165/166. See main text for details. (C) Basic Fuchsin staining of xylems in WT roots with longitudinal (upper panel) and radial (lower panel) views taken QC-proximally (upper panel) or QC-distally (lower panel) according to the picture in (D). Empty arrow heads: PX; filled arrow heads: MX. The graphs on the right-hand side depict the numbers of PX and MX files detected in $n = 30$ individual plants at the indicated positions. (D) Longitudinal view of a typical $A10^{NX}$ root, indicating the approximate positions used for sampling in panels (C) and (E–H) as well as in panels (A, B) and (D, E) of Fig. EV3. Scale bar: 200 µm. (E) same as (C) depicting the prevailing phenotypes of *ago10-1* roots. (F) Same as (C, E) depicting the phenotype of *phb phv cna* triple mutant roots in the *erecta 2* (*er-2*) background, used as a reference for normal MX-*vs*-PX differentiation. (G, H) As in (C, E, F) depicting the prevailing phenvos of *sdn1sdn2* double-mutant - (G) or $A10^{OX}$ (H) roots in $n = 22$ and $n = 30$ individual plants, respectively. See Dataset EV3 for the MX-*vs*-PX scores of individual plants used to produce the right-hand side graphs in panels (C, E, G, H). Data information: Error bars in (C, E, G, H): standard deviations. Scale bars in (C, E–H): 10 µm Source data are available online for this figure.

across an unspecified number of traversed cells (Fig. 5A), complicating the intertwined notions of competitive binding and gradient formation.

To establish the main mobile form(s) of miR165/166 in the RAM, we used layer-specific expression of a GFP-tagged allele of the P19 tombusviral silencing suppressor (GFP:P19). P19 specifically and cell-autonomously binds 21-22-nt sRNA duplexes, unlike longer ss/dsRNA (Brioudes et al, 2021; Brioudes et al, 2022; Brosnan et al, 2019; Devers et al, 2020; Vargason et al, 2003). We previously used stele-specific *pSHR::GFP:P19* (abbreviated *pSHR::G:P19*), endodermis-specific *pSCR::GFP:P19* (abbreviated *pSCR::G:P19*), and epidermis-specific *pWER::GFP:P19* (abbreviated *pWER::G:P19*) transgenic Arabidopsis (WT background; Fig. 5B) to physically capture, and thereby impede the activity of, certain mobile mi/siRNAs (Brosnan et al, 2019; Devers et al, 2020). If miR165/166 moved mainly as a processed, mature entity, it would be bound by GFP:P19 in both the *SCR* and *SHR* domains. This would, respectively, impede its movement from the miR165/166-incipient endodermis, and its activity inside the miR165/166-recipient stele. Both *pSHR::G:P19* and *pSCR::G:P19* roots should thus display related xylem development defects. If, conversely, pri-miR165/166 moved preponderantly from the endodermis to be subsequently processed and active inside the stele, only *pSHR::G:P19* roots should display xylem defects. Unlike the WT-like *pWER::G:P19* roots, both *pSHR::G:P19* and *pSCR::G:P19* roots displayed up to three additional MX files (Fig. 5B,C) evoking the *AGO10^{OX}* phenotype (Fig. 4H) although they had the cognate 2 PX files of WT roots. miR165/166 were detected in GFP:P19 IPs from dissected *pSHR::G:P19* and *pSCR::G:P19* whole root tips (Fig. 5D), collectively supporting the endodermis-to-stele movement of the processed, mature miR165/166. The correct PX specification in *pSHR::G:P19* and *pSCR::G:P19* (Fig. 5B,C) suggests that cell-specific P19 only incompletely suppresses mobile miR165/166 action in the endodermis and outer stele in which miR165/166 concentration is presumably highest. Abundant miRNAs can indeed outcompete P19's binding capacity (Brosnan et al, 2019; Devers et al, 2020). Moreover, enhanced degradation of AGO10-bound miR165/166 (Fig. 3B), presumably via SDN1/2 (Fig. 4G), is likely exacerbated in *AGO10^{OX}* yet would not affect the P19-bound fraction.

Most initials and the stele would receive the processed miR165/166 in the cytosol following their symplastic movement from the QC/CEI and endodermis, respectively (Fig. 2H). In both cases, the predominantly cytosolic signals for GFP:AGO1 and GFP:AGO10 (Fig. 1A) would concur with the proposed AGO10-*vs*-AGO1 competition for miR165/166 (Figs. 2G, 4B and 5A). The situation is distinct, however, in the AGO10-containing and

miR165/166-emitting CEI and QC (Fig. 2H). Unlike in all other cells receiving symplastic (i.e. cytosolic) miR165/166, pri-miR165/166 is probably transcribed, DCL1-processed and AGO1-loaded in the nucleus of the CEI and QC (Bologna et al, 2018) (Fig. 5A). Thus, AGO10 effects in the CEI and QC would likely entail a mainly nuclear AGO10-*vs*-AGO1 competition. Nuclear loading and subsequent export of AGO1:miRNA complexes in miRNA-producing cells is underpinned by AGO1 nucleo-cytosolic shuttling (Bologna et al, 2018; Zhang et al, 2020) (Fig. 5A). This property is evident upon cell treatment with leptomycin B (LMB), an inhibitor of the CRM1/XPO1-dependent export pathway causing GFP:AGO1 nuclear retention (Bologna et al, 2018). We confirmed that vacuum-infiltration allows LMB to penetrate the inner root tissues including the stele (where GFP:AGO10 and GFP:AGO1 overlap; Fig. 1A). Indeed, this treatment yielded nuclear signals in the stele of $pA1::G:A1^{(ago1-3)}$ (Fig. EV3F) as previously reported in the more accessible outer root layers (Bologna et al, 2018). Upon a similar treatment, the $AGO10^{NX}$ signal in the stele also became nuclear, supporting nucleo-cytosolic shuttling of AGO10 (Fig. 5E). Furthermore, transiently-expressed GFP:AGO10 displayed the same nucleo-cytosolic distribution as GFP:AGO1 and free eGFP in the *N.benthamiana* leaf epidermis (Fig. 5F). Therefore, the AGO10-*vs*-AGO1 competition for miR165/166 can occur in both cytosol and nucleus. The latter is possibly relevant within the *MIR165/166*-transcribing and miR165/166-emitting CEI/QC (Fig. 2H). An AGO10 nuclear pool would also accommodate partial pri-miR165/166 movement (Fig. 5A) and processing presumably in recipient cells' nuclei (Voinnet, 2022), a remaining possibility given the modest P19 effects on PX formation in the outer stele (Fig. 5B,C).

## Dynamic transcriptional and post-transcriptional regulations of PHB and AGO10 progressively restrict PHB accumulation along the xylem axis during early germination

Among five HD-ZIP III transcription factors (PHAVOLUTA or PHV; REVOLUTA or REV; PHB; CORONA or CNA; ATHB8), PHB is, genetically, key to xylem cell differentiation, with additional contributions from CNA and ATHB8 (reviewed in (Ramachandran et al, 2017)). Having identified AGO10 as a new player in this process, we investigated if and how AGO10's spatial distribution might contribute to form a *PHB* gradient from the outer (low levels, leading to PX formation) to the inner (high levels, leading to MX formation) cells of the stele (Fig. 4A). In line with

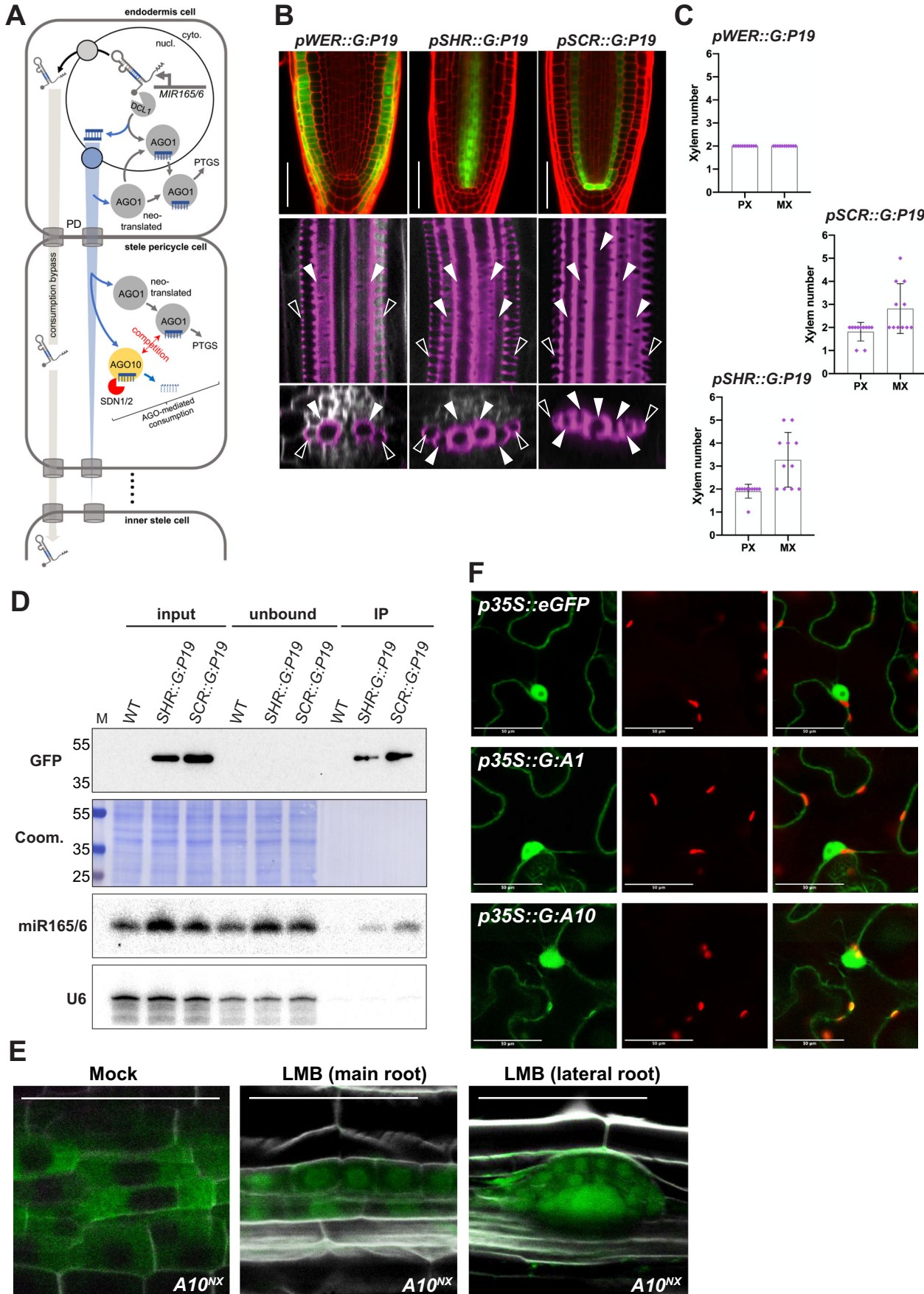

◀

**Figure 5. Subcellular localization of the AGO10-*vs*-AGO1 competition for mobile miR165/166 and possible mobile form of miR165/166.**

(A) Implications of its possible mobile forms (mature duplex *versus* pri-miRNA) to the non-cell autonomous action of miR165/166 from the endodermis, where it is exclusively produced, to the inner stele. PD: plasmodesma. (B) Upper panel: cell-autonomous, layer-specific accumulation of GFP:P19 in roots with the indicated genotypes. Empty arrow heads: PX; filled arrow heads: MX. (C) quantification of MX-*vs*-PX files in the genotypes depicted in (B) in *n* = 11 independent observations. Error bars: standard deviations. (D) GFP-based immunoprecipitation (IP) of P19:GFP in dissected whole root tips at 6 DAG in WT, *pSHR::G:P19* (SHR::P19) or *pSCR::G:P19* (SCR::P19) plants. Coom: Coomassie blue staining provides a total protein loading control. M: protein ladder with molecular weight in kDa. The indicated miRNAs were detected via hybridization of radiolabeled sequence-complementary oligonucleotides. Hybridization to the U6 snRNA provides a total small RNA loading control. The experiment was repeated three times with similar results. (E) Mock-*vs* vacuum-infiltrated LMB-treatments of *A10^NX* main or lateral roots. (F) Agrobacterium-mediated transient expression, under the *p35S* promoter, of the nucleo-cytosolic eGFP, GFP:AGO1 or GFP:AGO10 in *N.benthamiana* leaves. Data information: (B) cell walls were stained with PI yielding a red signal. Lower panel: Basic Fuchsin staining for xylems, and calcofluor white staining for cell walls in roots with the indicated genotypes. (E) See also Fig. EV3F for similar treatment of *pA1::GA1^(ago1-3)* roots. (F) Observations were made 6 days post-infiltration. All scale bars in (B), (E, F): 50 µm. Source data are available online for this figure.

our root tips' observations (Fig. 2H), such a gradient was previously proposed to explain cognate cell-type specification along the xylem axis (Carlsbecker et al, 2010; Miyashima et al, 2011). We conducted a parallel time-course analysis of all major protagonists from 3 to 6 days-after-germination (DAG). Unlike in the experiments depicted so far, projections of the longitudinal confocal views were made so as to reveal the stele's inner parts and the xylem axis in particular. Reconstructed orthogonal projections provided radial plan observations of the same regions.

At 3 DAG, the transcriptional signal from *pA10::NTF:G*#1 was widespread throughout the stele's QC-proximal region, albeit less pronounced in the pericycle; its intensity then progressively decreased from 4-to-6 DAG, remaining mostly unchanged thereafter (Fig. 6A). The translational signal from *A10^NX* was similarly decreased, but a disproportionate loss-of-GFP:AGO10 accumulation was uniquely observed along a region apparently delineating the xylem axis (Fig. 6A). The suggested spatially-restricted post-transcriptional down-regulation of the *AGO10*/AGO10 mRNA/protein was strikingly rapid. Unnoticeable at 3 DAG, it was clearly visible at 4 DAG (Fig. 6A). Moreover, *A10^NX* yielded only a very weak signal at 1 DAG (Fig. EV4A), suggesting that the bulk of AGO10 is produced de novo during early germination. At 5–6 DAG, the *pA10::NTF:G*#1 signal also started to decrease, yet more prominently so along the same region (Fig. 6A, arrows). Nonetheless, transcription was never abolished, whereas the GFP:AGO10 protein was already below detection in this region (xylem axis).

Imaging calcofluor-treated *A10^OX* root tips (to enhance the signal) at 6 DAG confirmed that GFP:AGO10 is depleted precisely along the xylem axis, resulting in a pincer-like pattern reflecting its accumulation in the procambium and phloem on each axis' side; the pincer is surrounded by the stele's pericycle containing substantially lower albeit detectable GFP:AGO10 levels (Fig. 6B). Longitudinally, the xylem-axis depletion of GFP:AGO10 was manifested across its entire expression domain in the stele, from the very first cells located just above the QC and pro-vascular initials (10-25 µm; Fig. 6B). Thus, accompanying a rapid and substantial decrease in the overall *AGO10*/AGO10 transcription/protein levels from 3-to-6 DAG, combined transcriptional and post-transcriptional mechanisms further spatially refine AGO10 accumulation inside the stele to form a pincer therein. Confirming the microscopy observations, endoAGO10 accumulation showed a sharp deficit at 5-8 DAG in western blot analysis conducted in whole non-transgenic WT root tips; endoAGO1 levels remained, by contrast, essentially unchanged (Fig. EV4B). Consistent with the AGO10-bound miR165/166 fraction undergoing degradation presumably by SDN1/2 (Figs. 3B and 4G), the total levels of miR165/166, but not of miR158, were inversely

correlated with those of endoAGO10 along the germination time-course (Fig. EV4C).

At 3–4 DAG, the *pPHB::2B:G* transcriptional reporter yielded a signal within the stele that extended toward the surrounding ground tissue and beyond. By contrast, the signal from the *pPHB::PHB:G* translational reporter was restricted to the stele where, undetectable in the outer pericycle, it appeared gradually more pronounced in the inner part (Fig. 6C), as reported (Miyashima et al, 2011). These observations support the notion that AGO10, by being prominently expressed in the inner stele, protects *PHB* therein against AGO1-dependent silencing caused by a gradient of mobile miR165/166 originating from the endodermis and affecting all surrounding layers. Accordingly, *pA1::G:A1^(ago1-3)* yielded a similar signal in all root layers with an unchanged intensity from 3-to-6 DAG, as was *MIR165a/MIR166b* transcription (Figs. EV4B and 5D). Remarkably, in several independent transgenic lines, the shape of the graded signal yielded by *pPHB::PHB:G* changed over time. Mostly radial at 3–4 DAG, the gradient became progressively focused along the xylem axis from 5-to-6 DAG in a manner apparently coordinated with the formation of the GFP:AGO10 pincer (Figs. 6C and EV5A). At 8 DAG, the PHB:GFP signal was precisely encased within the pincer and mostly apparent in the 2-3 MX cells in the center of the xylem axis, whereas it was low in each PX cell at its periphery (Fig. 6D), as indeed anticipated in the model in Fig. 4A. Imaging along the root tip's longitudinal axis at 8 DAG revealed that the MX-focused PHB:GFP signal and that of the surrounding GFP:AGO10 pincer were coordinately restricted to the tip's QC-proximal region. By contrast, the transcriptional signals from *pPHB::2B:G* and from the functionally-redundant *MIR165a/MIR166b* were, respectively, widespread in the stele and endodermis-specific along the entire length of the root tip (Fig. 6E). None of the signals analyzed evolved further after 8 DAG.

The intensity of the *pPHB::PHB:G* signal, as opposed to its shape, did not appreciably change during early germination points of 3-to-6 DAG (Fig. 6C). This was surprising because the levels of GFP:AGO10—proposed here to protect *HD-ZIPIIIs* against AGO1-induced PTGS—were substantially decreased from 5 DAG onward (Fig. 6A). A likely explanation was provided by the much stronger signal yielded by the *pPHB::2B:G* transcriptional reporter at 5-to-6 compared to 3-to-4 DAG (Fig. 6C). Thus, a substantial gain in *PHB* transcript levels presumably compensated for the reduced AGO10 levels at these later time points. We expressed the *PHB::PHB:G* transgene in the *ago10-1* background. In four independent lines analyzed, the PHB:GFP signal was strongly reduced compared to that observed in the WT background (Fig. 6C). Moreover, applying

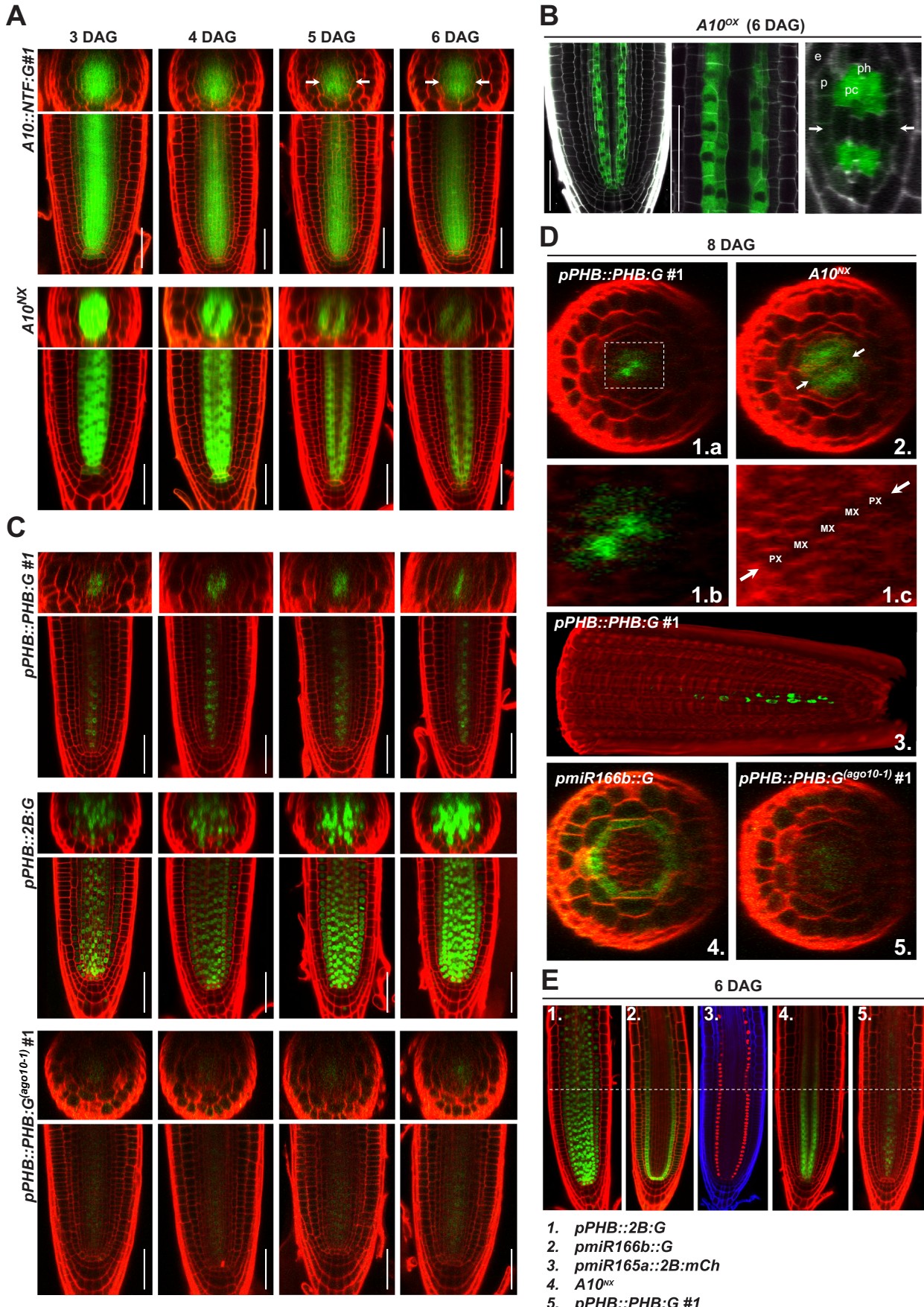

1.  pPHB::2B:G
2.  pmiR166b::G
3.  pmiR165a::2B:mCh
4.  A10ᴺˣ
5.  pPHB::PHB:G #1

◄

**Figure 6.   An AGO10 pincer simultaneously quenches the excess of miR165/166 and focuses its movement along the xylem axis.**

(A) Three-to-six DAG time-course analysis of the GFP signals yielded by the *A10^NX* translational reporter *vs A10::NTF:G* transcriptional reporter in the main root, with radial (upper panels) and longitudinal (lower panels) views. The arrows in *A10::NTF:G* indicate stronger signal depletion along the xylem axis starting at 5 DAG. (B) Calcofluor white-treated main root tips *of A10^OX* at 6 DAG. e: epidermis, p: pericycle, ph: phloem, pc: procambium. (C) Same as in (A) for: (1)-top panel: *pPHB::PHB:G* translational reporter line #1 in the WT background (upper panel) out of 4 independent T2 lines (see also Fig. EV5A,B), as used in panel 4 of Fig. 2H; (2)-middle panel: reference *PHB::2B:G* transcriptional reporter T3 line also used in panel 3 of Fig. 2H; or, (3)-bottom panel: *pPHB::PHB:G* translational reporter line #1 in the *ago10-1* background (upper panel) out of 4 independent T2 lines (see also Fig. EV5A,B), as used in panel 6 of Fig. 2H. (D) GFP signals yielded by the indicated reporters (as used here in panels (A) and (C)) in the main root at 8 DAG. Panel 1.b is an excerpt of panel 1.a; the enhanced-contrast panel 1.c is a red-channel-only view of panel 1.b allowing MX and PX visualization. Note that the nucleus from the third MX in panel 1.b was not in the same plan as the nuclei of the two adjacent MX cells, hence the absence of a *pPHB::PHB:G* signal therein. Panel 3. is a 3D reconstruction from an 8 DAG root similar to that observed in panel 1a. Arrows: as in (B). (E) Longitudinal views of 6 DAG roots expressing the indicated reporters. The dashed line marks the end-point of *A10^NX* expression, which nearly coincides with that of *pPHB::PHB:G*, consistent with a *PHB*-protective role for AGO10 against miR165/166-mediated silencing. Source data are available online for this figure.

high contrast using the FIJI post-processing software indicated that the low signal remaining in *ago10-1* did not become progressively focused along the xylem axis over the 3-to-6 DAG time course analysis, unlike in WT (Fig. EV5B). Instead, it remained distributed throughout the inner stele, as observed in WT roots at 3–4 DAG (Fig. EV5B). Collectively, these results suggest that AGO10 not only quenches *PHB* silencing caused by mobile miR165/166, but also simultaneously helps focus the *PHB* gradient along the xylem axis during early germination.

## A refined model for *HD-ZIP III* silencing and xylem development in the Arabidopsis root

Our time-course analysis leads us to distinguish two types of AGO10 contributions considering that AGO1 and miR165/166 levels remain mostly constant over time (Fig. EV4B–D). During early germination and up to 4 DAG, AGO10 likely competes strongly for miR165/166 binding in the stele, in which AGO10 expression is high and homogenous, save in the pericycle. Consumption of mobile miR165/166 by AGO10 via binding and SDN1/2-mediated turnover would confer substantial protection to the moderately-expressed *PHB* mRNA. Since miR165/166 should undergo dilution as it moves away from presumably every cell of the endodermis, protection by AGO10 would be highest in the central part of the stele and lowest in its outer part, resulting altogether in a radial *PHB* gradient. However, from 5 DAG onward, AGO10 levels are substantially reduced and a pincer-like structure is established in the stele around the xylem axis. At this time, *PHB* transcription is significantly enhanced. Given the contemporary AGO10 deficit, the ensuing abundant transcripts would now be more available for AGO1:miR165/166-dependent PTGS, resulting in robust target-directed miRNA degradation (TDMD; Shi et al, 2020) of mobile miR165/166. We propose that from 5 DAG onward, AGO10 plays a mere buffering function by quenching the remaining miR165/166 pool not consumed by AGO1-mediated TDMD. Quenching would preclude further ingress of this remaining pool inside the stele, except in the AGO10-free xylem axis in which the *PHB* gradient would be thereby primarily established but with only a fraction of the total miR165/166 emitted from the surrounding endodermis (Figs. 2H, EV2A and 6D,E). *HD-ZIP III* silencing would be predictably stronger under the xylem pole pericycle, promoting PX, and less intense in the central part, promoting MX (Figs. 4A and 7).

The ensuing refined model (Fig. 7) suggests how inversely-correlated miR165/166-*vs*-HD-ZIP III gradients might form precisely along the xylem axis despite (i) miR165/166 moving from

theoretically every endodermis cell surrounding the root vascular cylinder and (ii) transcription of *PHB/CNA* (but not *ATHB8*) being widespread in the whole stele (Carlsbecker et al, 2010; Miyashima et al, 2011; Ramachandran et al, 2017) (Fig. 6C), as is accumulation of miR165/166-resistant alleles of *PHB:GFP* (Carlsbecker et al, 2010; Miyashima et al, 2011). The suggested AGO10-orchestrated buffering and channeling of mobile miR165/166 within the stele further clarifies several questions left unaddressed in a previous model for xylem differentiation which did not involve AGO10. First, it possibly explains why *in situ* hybridizations detect miR165/166 very poorly, if at all, inside the stele (Carlsbecker et al, 2010; Fan et al, 2021), whereas an endodermis->cortex->epidermis gradient can be observed (Carlsbecker et al, 2010). It was hypothesized that in situ RNA probes might only access a free, i.e. AGO1-unbound miR165/166 pool, such that strong AGO1-mediated miR165/166 consumption via *HD-ZIP III* silencing would hinder their detection in the stele (Carlsbecker et al, 2010). However, the signals for AGO1-loaded miR165/166 and *HD-ZIP III* translatomes are comparable in the stele, ground tissue, and epidermis (Fig. EV4E). Since the procambium, phloem, and to a lower extent, pericycle, form together the main bulk of the stele's proper, we propose that miR165/166 degradation coupled to AGO10 expression therein (Fig. 6A,B) likely underlies their global lack-of-detection. The remaining, small fraction of mobile miR165/166 available for *HD-ZIP III* gradient's formation along the xylem axis (Fig. 7) would probably be too low for robust in situ detection. A second aspect pertains to somewhat varying accounts of the *PHB* gradient's shape within the stele. Predominantly radial in some studies (Carlsbecker et al, 2010; Miyashima et al, 2011) as indeed observed here at 3–4 DAG, it appears mainly restricted to the xylem axis in others (Fan et al, 2021), as observed at 5 DAG and beyond (Fig. 6C). We suggest that the final shape of the gradient is established during early germination by adjusting both the level and spatial position of AGO10 on the one hand, and *PHB* transcription levels on the other (Fig. 6A–C). Given the highly dynamic nature of these processes over just a few days, the timing of sampling/observation possibly explains and reconciles these previous differences.

## Concluding remarks

The present study identifies *hitherto* unknown functions for AGO10 in the Arabidopsis RAM, namely the control of meristem activity/length likely in the layers' initials, and of PX-*vs*-MX development in the stele. El Arbi et al (2024) additionally show, in

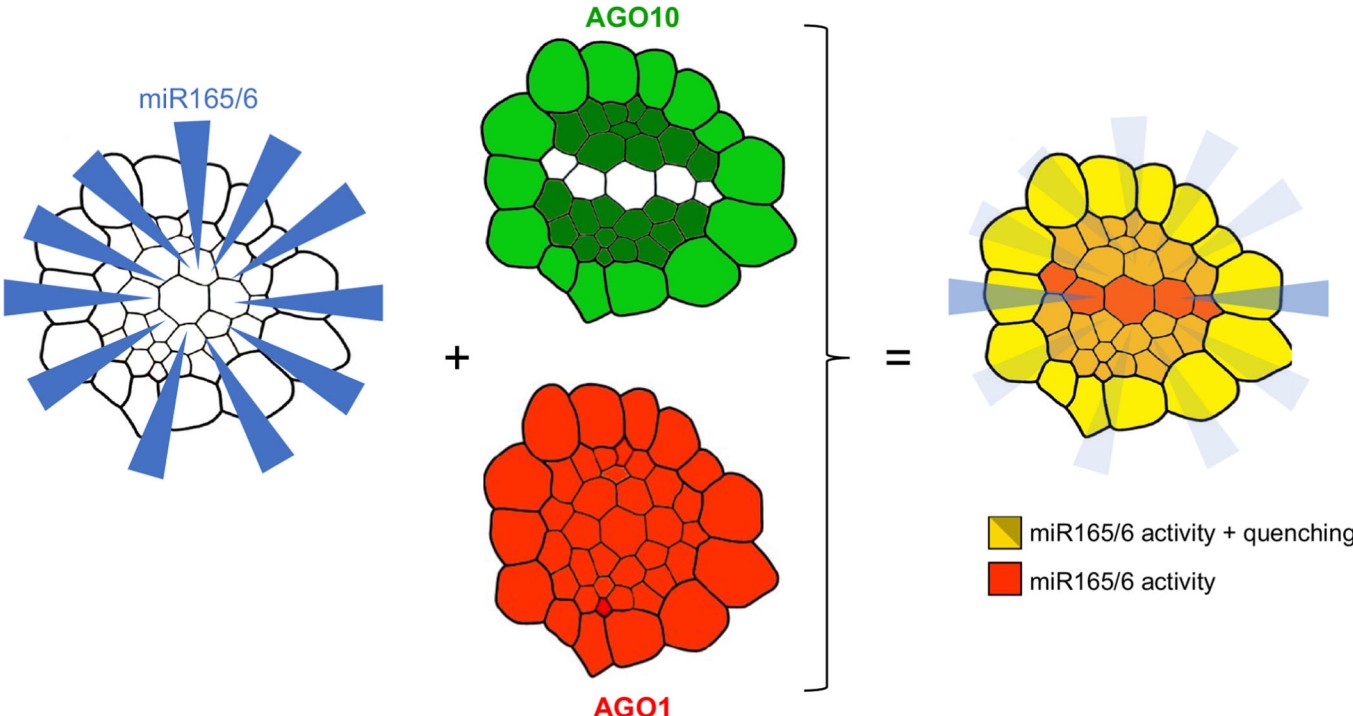

**Figure 7. A refined model for *HD-ZIP III* silencing and xylem cell differentiation from 5 DAG onward.**

See main text for details.

the accompanying study, that AGO10 buffers the vascular cytokinin response required for formative divisions in the root vasculature. While the extent to which they are mechanistically interconnected remains to be formally established, all these functions expectedly involve quenching of mobile miR165/166 activity and, hence, modulation of AGO1-mediated *HD-ZIP III* silencing. This modulation is possibly granted by the dynamic spatio/temporal control of AGO10 accumulation, of which a substantial part seems post-transcriptional. This was particularly evident along the stele's longitudinal axis and within its radial plan, in the division zone, during the AGO10 pincer formation. How such post-transcriptional control might be operated and directed to act precisely along the xylem axis is a fascinating prospect for future studies, as is the examination of the suggested AGO10-mediated modulation of the initials' activity. Regardless of the mechanisms involved, one foreseeable and key underpinning of post-transcriptional control is an ability to confer versatility and robustness to AGO10-regulated processes, by enabling rapid changes and/or fine-tuning of AGO10 levels in response to environmental cues/stress. The ensuing potentiation of phenotypic plasticity is well illustrated by El Arbi et al (2024) under water deficiency. Drought-induced abscisic acid (ABA) signaling indeed increases miR165/166 levels and possibly decreases those of AGO10 in the root, likely contributing to dynamic vascular acclimation through altered PX-*vs*-MX specification (Bloch et al, 2019; Ramachandran et al, 2020; Ramachandran et al, 2018). Our study also provides compelling support to the previously-made suggestion that strategic positioning of AGO proteins—which "consume" mobile mi/siRNAs as they move away from their production sites—might empower a large diversity of spatial sRNA activity patterns

(Devers et al, 2020; Voinnet, 2022). This is advocated here with the stele's AGO10 pincer, which probably simultaneously quenches (in the pericycle, phloem, and procambium) and focuses (along the xylem axis) the radial signaling process endowed by the abundant mobile miR165/166 produced in the surrounding endodermis (Fig. 7).

## Methods

### Plant material and growth conditions

All *Arabidopsis thaliana* lines used in this study were in the Col-0 ecotype background. Unless specified in the nomenclature of the lines (upper case, parenthesis), the genotype is wild-type (WT). The *ago10-1* null mutant (T-DNA insertion line), *shr-2* mutant, and *sdn1sdn2* double mutant were described, respectively, in (Takeda et al, 2008), (Carlsbecker et al, 2010), and (Ramachandran and Chen, 2008). The *pA10::G:A10*(ago10-1) T2 lines used for selecting T3 lines *A10*[UX], *A10*[OX,] and *A10*[NX] used throughout this study were described in (Iki et al, 2018; Jullien et al, 2022). Line *pA1::G:A1*(ago1-3) was described in (Bologna et al, 2018). Lines *pA10::NTF:G#1/2* are from (Palovaara et al, 2017). Lines *pWER::G:P19*, *pSHR::G:P19*, and *pSCR::G:P19* were described in (Brosnan et al, 2019; Devers et al, 2020). Lines *pMIR166b::G* and *pMIR165a::H2B:mch* were described in (Brosnan et al, 2019). For plant growth, surface sterilized seeds were sown on ½MS medium containing MES buffer (Duchefa Biochemie) and solidified with 0.8% plant-agar with no sucrose. Seeds were vernalized at 4 °C for 48 h upon which germination and growth was conducted under short day conditions at 22 °C on

vertical plates. Typically, 6 DAG seedlings were used for observations and molecular analysis unless otherwise specified.

## Cloning procedures and plant transformation

The *pPHB::H2B:GFP* and *pAGO10::H2B:GFP* expression vectors were cloned using the multi-site gateway system (Invitrogen). The *pPHB* and *pAGO10* promoter sequences were PCR-amplified from Arabidopsis WT genomic DNA (primer sequence available upon request) and recombined into the Gateway vector pDONR4-1r. The afore mentioned promoter- and eGFP entry-clones (in pDONR P2R-P3; (Pumplin et al, 2016)) were then recombined with an H2B entry clone (Jullien et al, 2012) into the destination vector pB7m34GW (Karimi et al, 2007). T2 lines *pPHB::PHB:G* #1-4 and *pPHB::PHB:G(ago10-1)* #1-4 were engineered using the transformation vector described in (Miyashima et al, 2011) kindly provided to us by Dr. S. Miyashima (Nara Institute of Science and Technology, Japan). Kanamycin-resistant lines *pPHB::PHB:G* #1-4 were further selected based on their GFP expression pattern, similar to that reported in (Miyashima et al, 2011) and on their normal (2 PX)-(2 MX) xylem phenotype in the root QC-distal region. By contrast, most GFP-negative T2s had no MX and multiple PX therein, suggesting co-suppression of *HD-ZIP IIIs* via transitive silencing initiated by mR165/166 against the *pPHB::PHB:G* mRNA. Kanamycin-resistant lines *pPHB::PHB:G(ago10-1)* #1-4 were selected based on low, yet detectable GFP accumulation in the stele, and on the typical (4 PX)-(2 MX) phenotype consistently observed in the QC-distal region of *ago10-1* roots, as shown in Fig. 4E, lower panel. All constructs were introduced into Arabidopsis (ecotype Col-0) using the floral dip method (Clough and Bent, 1998).

## Transient expression

Transient expression in *N.benthamiana* leaves was performed as described (Bologna et al, 2018). For the experiments in Fig. EV1D, Agrobacterium cells were diluted to a final optical density of 0.3 and samples were collected 2 days post-infiltration. *p35S::GFP:AGO1 (p35S::G:A1)* and *p35S::eGFP* were as described (Bologna et al, 2018; Brioudes et al, 2022; Jay et al, 2023). *p35S::GFP:AGO10 (p35S::G:A10)* was obtained by swapping the *pAGO10* promoter for *p35S* from construct *pA10::G:A10* (Iki et al, 2018; Jullien et al, 2022).

## RNA extraction and northern analysis

RNA was extracted from frozen, dissected whole root tips (~8 mm from the tips) ground in liquid nitrogen using TRI Reagent (Merck) according to the manufacturer's instructions and resuspended in water. Equal amounts of RNA (1 to 10 μg), dried with a vacuum concentrator, or immunoprecipitated RNA fractions, were resuspended in 50% formamide northern blot loading buffer and resolved by electrophoresis in 0.5XTBE on a 17.5% denaturing polyacrylamide gel containing 8 M urea. RNA was transferred on a Hybond-NX Nylon membrane (Merck Sigma) in 0.5X TBE, and cross-linked using 1-ethyl-3-(3-dimethylaminopropyl)carbodiimide (EDC), according to (Pall and Hamilton, 2008), for 2 h at 60 °C. Membranes were incubated overnight in PerfectHyb Plus Hybridization buffer (Merck Sigma) at 42 °C, with an oligonucleotide probe 5'-end labeled with [γ-$^{32}$P]ATP using T4 PNK (Thermo

Fisher Scientific) and complementary to the specified miRNA sequence. Membranes were washed 3 times with 2X SSC, 2% SDS at 50 °C for 15 min. The membrane was then exposed over night to a phosphor screen (Fuji) and scanned using a Typhoon FLA 9000 scanner (GE Healthcare). Multiple sequences were probed on individual membranes by stripping twice with boiling 0.1% SDS for 15 min before re-probing.

## Protein extraction and western analysis

Total proteins were extracted from frozen, dissected whole root tips by grinding in liquid nitrogen and resuspending in two volumes of RIPA buffer (50 mM Tris, 150 mM NaCl, 1% NP-40, 0.5% sodium deoxycholate, 0.1% sodium dodecyl sulfate [pH 7.5]) containing Roche Complete protease inhibitors. Extracts were cleared by centrifugation at 12 K for 10 min at 4 °C. Protein concentrations were normalized using a modified Lowry procedure with the DCTM Protein Assay Kit (Bio-Rad), resolved on SDS-PAGE gels, and electro-transferred to Immobilon-P PVDF membranes (Millipore). After blocking for 30 min in 1X PBS + 0.1% Tween-20 supplemented with 5% skimmed milk powder, subsequent antibody incubations were carried out overnight at 4 °C in the same solution. Primary anti-AGO1 (Agrisera AS09 527), anti-AGO10 (Grentzinger et al, 2020)) or anti-GFP (Chromotek 3H9) antibodies were, respectively, diluted at 1/8000, 1/1000 or 1/5000. Membranes were washed 3 times with PBS-T (1X PBS + 0.1% Tween-20) and incubated for 1 h at room temperature with 1:10,000 dilutions of HRP-conjugated goat anti-rat (GFP western analysis) secondary antibody (Cell Signalling, ref. #7077s) or HRP-conjugated goat anti-rabbit (AGO1 and AGO10 western analysis) secondary antibody (Thermo Fisher Scientific, ref. 65-6120) in PBS-T 5% milk and washed again 3 times with PBS-T. Protein detection was carried out with the Westar Supernova ECL substrate (Cyanagen) and imaged via the ChemiDoc Touch Imaging System (Bio-Rad) with the auto-exposure setting. Membranes were stained with Coomassie blue to reveal total protein.

## Immunoprecipitation

Frozen, dissected whole root tips ground in liquid nitrogen were resuspended in 1 mL (for 200 g of powder) IP buffer (50 mM Tris-HCl pH 7.5, 150 mM NaCl, 10% glycerol, 0.1% NP40), containing 2 μM MG-132 and one tablet of cOmplete® protease inhibitor cocktail (Merck Roche) per 10 mL. After 30 min mixing, lysates were cleared from cell debris twice by centrifugation at $10,000 \times g$ for 15 min. 100 μL of cleared supernatants were mixed to 4X western blot loading buffer for analysis of input protein fractions and 100 μL were collected for RNA extraction. For GFP:AGO1, GFP:AGO10, and GFP:P19 IPs, lysates were first pre-cleared with 40 μL of agarose beads (Merck Roche) for 1 h on a rotating wheel at 4 °C. Pre-cleared lysates were then incubated for 1 h on a rotating wheel with 30 μL of GFP-trap magnetic agarose beads (Chromotek), pre-blocked with 2% BSA in IP buffer. Agarose or magnetic bead conjugates were washed 4 times with IP buffer for 10 min, collected, and resuspended in 500 μL of TRI Reagent (Merck) for RNA extraction according to the manufacturer's instructions, or 500 μL of acetone for protein extraction upon which RNA or protein detection were conducted as described above. For endoAGO1, lysates were first pre-cleared with 30 μL of protein-A-agarose beads (Merck Roche) for 30 min on a

rotating wheel at 4 °C. Pre-cleared lysates were then incubated for 2 h on a rotating wheel with anti-AGO1 antibody from Agrisera (1:800), followed by an incubation of 1 h with 30 μL of protein-A-agarose beads (Merck Roche) at 4 °C. Agarose beads were washed 4 times with IP buffer for 10 min. 20% of the beads were mixed to 4X western blot loading buffer for analysis of IP protein fractions and 80% of the beads were resuspended in 1 mL of TRI Reagent (Merck) for RNA extraction according to the manufacturer's instructions.

## Confocal imaging

Fluorescent transcriptional or translational fusion-expressing roots were imaged under a Zeiss 780 confocal laser scanning microscope using 40x water immersion objectives immediately after their excision from growing seedlings following propidium iodide cell wall staining (50/1000 μl dilution in water). For histological staining (calcofluor or basic Fuchsin), seedlings were fixed in 4% PFA (paraformaldehyde) in 1X PBS for 1 h with gentle mixing. Seedlings were then briefly washed twice in 1X PBS before being cleared by ClearSee (Kurihara et al, 2015) (10% xylitol, 15% sodium deoxycholate, 25% urea) for at least 24 h. To visualize the cell wall, seedlings were incubated in 0.1% calcofluor white (diluted in ClearSee) for 30-to-60 min For lignin staining of xylem files, seedlings were incubated in 0.2% basic Fuchsin (diluted in ClearSee) overnight. GFP fluorescence was excited with the 488 nm laser line and its emission collected between 520–550 nm. mCherry fluorescence was excited with the 561 nm laser line and its emission collected between 575–650 nm. Basic Fuchsin fluorescence was excited with the 561 nm laser line and its emission collected between 600–650 nm. Calcofluor white fluorescence was excited with the 405 nm laser line and its emission collected between 425–475 nm. Propidium iodide fluorescence was excited with the 488 nm laser line and its emission collected at 615 nm. For radial cross sections, an orthogonal view of the root was generated using the stacking function of the Zeiss 780 software. For the 3D reconstruction in Fig. 6D (panel #3), the Zeiss software was used to digitally compile all the acquired stacks and yield a 3D projection of the root. To calculate gradient steepness of *pA10:NTF:G #1* versus *A10^{NX}* (Fig. 1C; Dataset EV1), we determined the total fluorescence of a signal by subtracting out the background signal, within an identical sized section at various positions towards the QC (5, 70, and 150 μm), using Fiji software. The corrected total cell fluorescence (CTCF) was calculated using this formula: CTCF = integrated density – (area of selected cell × mean fluorescence of background reading). Images shown in figures are representative of consistent results observed in multiple experiments, as indicated in the figures' legends.

## LMB treatment

LMB treatment of roots was adapted from (Bologna et al, 2018). A vacuum of −0.05 MPa was applied for 30 min to whole seedlings suspended in the LMB buffer. Seedlings were then incubated without vacuum for a further 30–60 min before fixation with 4% PFA followed by ClearSee clearing and Calcofluor white staining. Seedlings were then inspected under confocal microscope.

## Data availability

This study includes data deposited on Imaging dataset BioImages under the accession number S-BIAD968.

## Peer review information

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

## Acknowledgements

We thank Voinnet lab members, Dr. Bojan Gujas and Dr. Emanuel Devers for discussions and critical reading of the manuscript and the Scientific Center for Optical and Electron Microscopy (ScopeM) of the ETH-Z (SCOPE-M) for support. We thank Dr. Emanuel Devers and Peiqi Lim for the *pSHR::2B:G* and *pPHB::2B:G* seeds, respectively. SM was supported by a Swiss Government Excellence post-doctoral Scholarship (no. 2018.0855) and a post-doctoral grant from the ETH-Z (no. ETH-32 19-2). Aspects of this work were supported by a research grant from the Swiss National Foundation (SNF) no. 310030 197832 "*Genome-scale, functional and mechanistic investigation of plant microRNA cell-to-cell and vascular movement*".

## Author contributions

**Shirin Mirlohi**: Conceptualization; Resources; Data curation; Software; Formal analysis; Funding acquisition; Validation; Investigation; Visualization; Methodology; Writing—original draft; Project administration; Writing—review and editing. **Gregory Schott**: Data curation; Validation; Gregory Schott has helped with molecular analysis. **Andre Imboden**: Resources; Data curation; Andre Imboden helped perform crosses and genotyping as well as growing the plants used in the study. **Olivier Voinnet**: Conceptualization; Supervision; Funding acquisition; Writing—original draft; Project administration; Writing—review and editing.

## Disclosure and competing interests statement

The authors declare no competing interests.

# Expanded View Figures

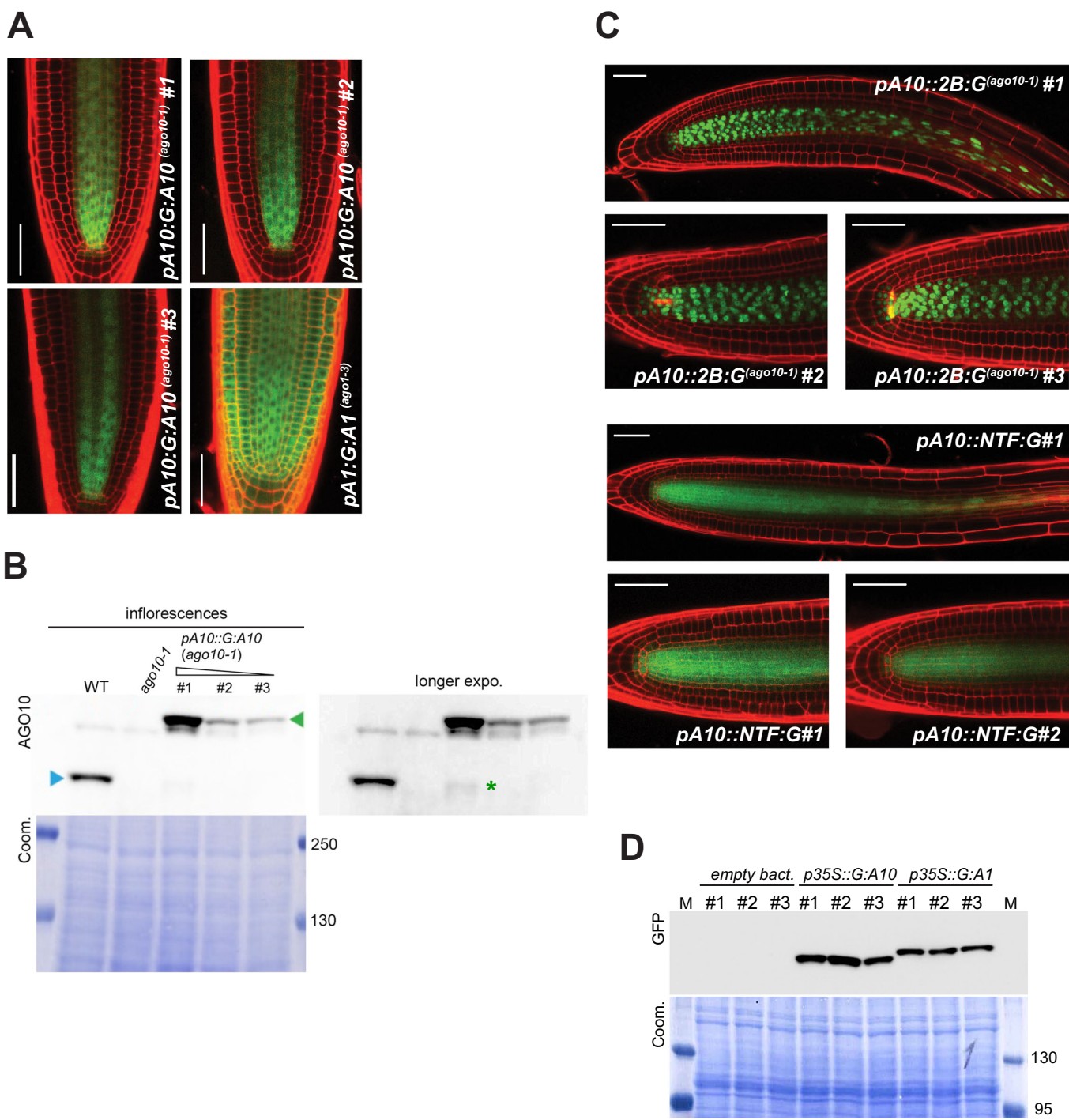

**Figure EV1.   AGO10 expression and transgenic lines' characterisation.**

(A) QC-proximal views of the GFP signals yielded in root tips by the indicated reporters in the indicated genetic backgrounds. (B) Western analyses of GFP:AGO10 (green arrow) and endoAGO10 (blue arrow) accumulation in inflorescences where the anti-AGO10 antibody yields almost no non-specific or background signals. The longer exposure on the right-hand side allows detection of a GFP::AGO10-derived degradation fragment in overexpressor line #1, indicated with a green asterisk. Coom: Coomassie blue staining provides a total protein loading control. M: protein ladder with molecular weight in kDa. (C) GFP signals yielded in root tips by the indicated reporters in the indicated genetic backgrounds. All scale bars: 50 μm. cell walls were stained with PI yielding a red signal. (D) Transient expression of *p35S::G:A10* or *p35S::G:A1* in *N.benthamiana* leaves. Samples were collected 2 days post-infiltration and the extracted proteins subjected to western analysis using an anti-GFP antibody. Both proteins accumulate similarly. Coom: Coomassie blue staining provides a total protein loading control. M: protein ladder with molecular weight in kDa. Data information: (A, C) Roots were inspected at 6 DAG. Source data are available online for this figure.

**A**

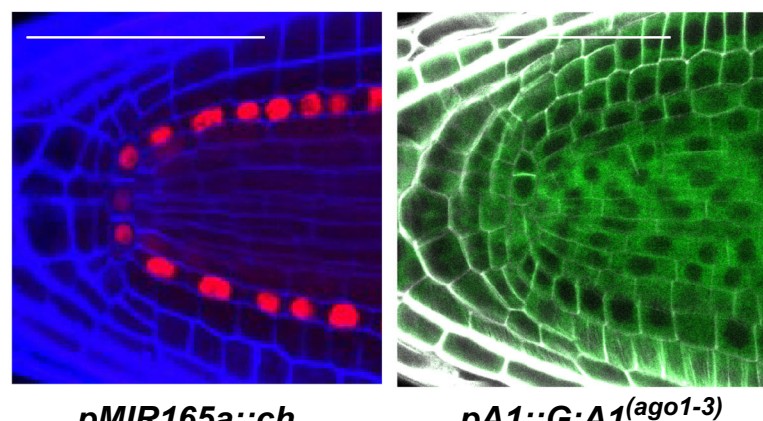

*pMIR165a::ch*            *pA1::G:A1*$^{(ago1-3)}$

**B**

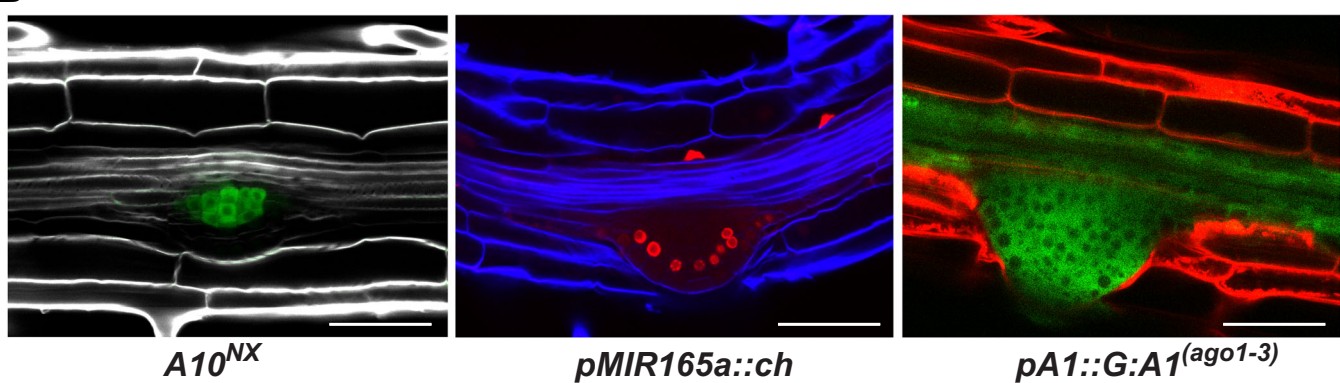

*A10*$^{NX}$            *pMIR165a::ch*            *pA1::G:A1*$^{(ago1-3)}$

**Figure EV2.  *MIR165a* transcription and GFP-AGO1 accumulation in the root tip.**

(**A**) QC-proximal views of the mCherry and GFP signals yielded in 6 DAG root tips of, respectively, the *pMIR165a::ch* transcriptional- and *pA1:G::A1*$^{(ago1-3)}$ translational-reporter lines. (**B**) GFP or mCherry signals yielded by the indicated reporters in lateral root initiation sites in the indicated backgrounds. Cell walls were stained by calcofluor white (in white) or PI (in red). Scale bars in (**A, B**): 50 µm. Source data are available online for this figure.

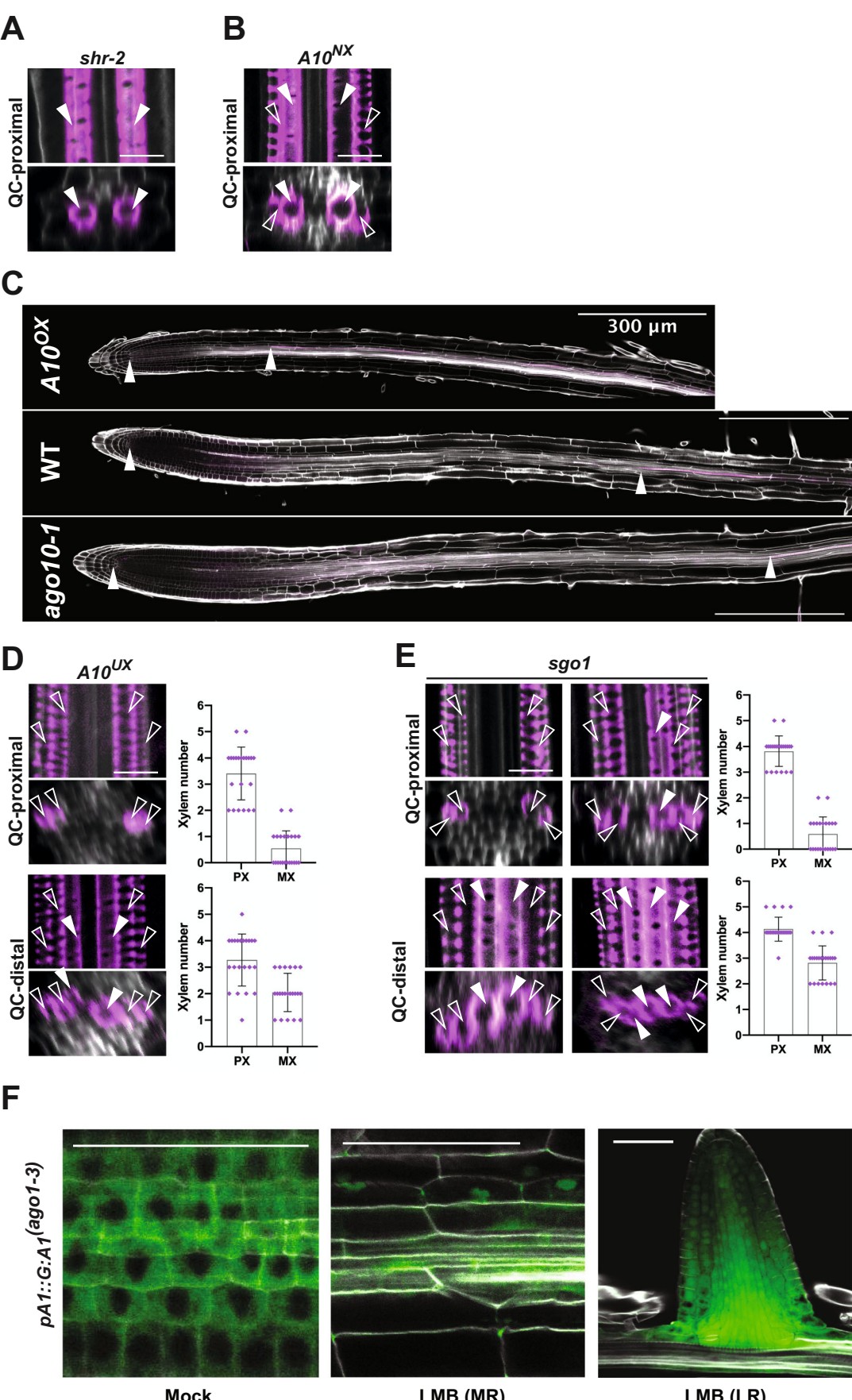

◀

**Figure EV3.    Basic Fuchsin staining and meristem length in the indicated genotypes as well as LMB treatments.**

(**A, B**) Basic Fuchsin staining of the differentiated xylem in the QC-proximal regions of the indicated genotypes. Empty arrows: PX; filled arrows: MX. Scale bars: 10 µm. (**C**) Basic Fuchsin staining to reveal the distance between the QC (left arrows) and the initiation of mature xylem cells (having gained their secondary cell wall; right arrows) in the indicated genotypes. Scale bars: 300 µm. (**D, E**) Basic Fuchsin staining of xylems in both QC-proximal (upper panels) and QC-distal (lower panels) root regions. The graphs on the right-hand side depict the numbers of PX and MX files detected in $n = 22$ plants at the indicated positions. Empty arrow heads: PX; filled arrow heads: MX. Scale bars: 10 µm. See Dataset EV3 for the MX-*vs*-PX scores of individual plants used to produce these graphs. Error bars: standard deviations. (**F**) Mock or vacuum-infiltrated LMB treatments within the stele of *pA1::G:A1*$^{(ago1-3)}$ main roots (MR) or lateral roots (LR). Note GFP:AGO1 relocalization from the cytosol to the nucleus upon LMB treatment. Source data are available online for this figure.

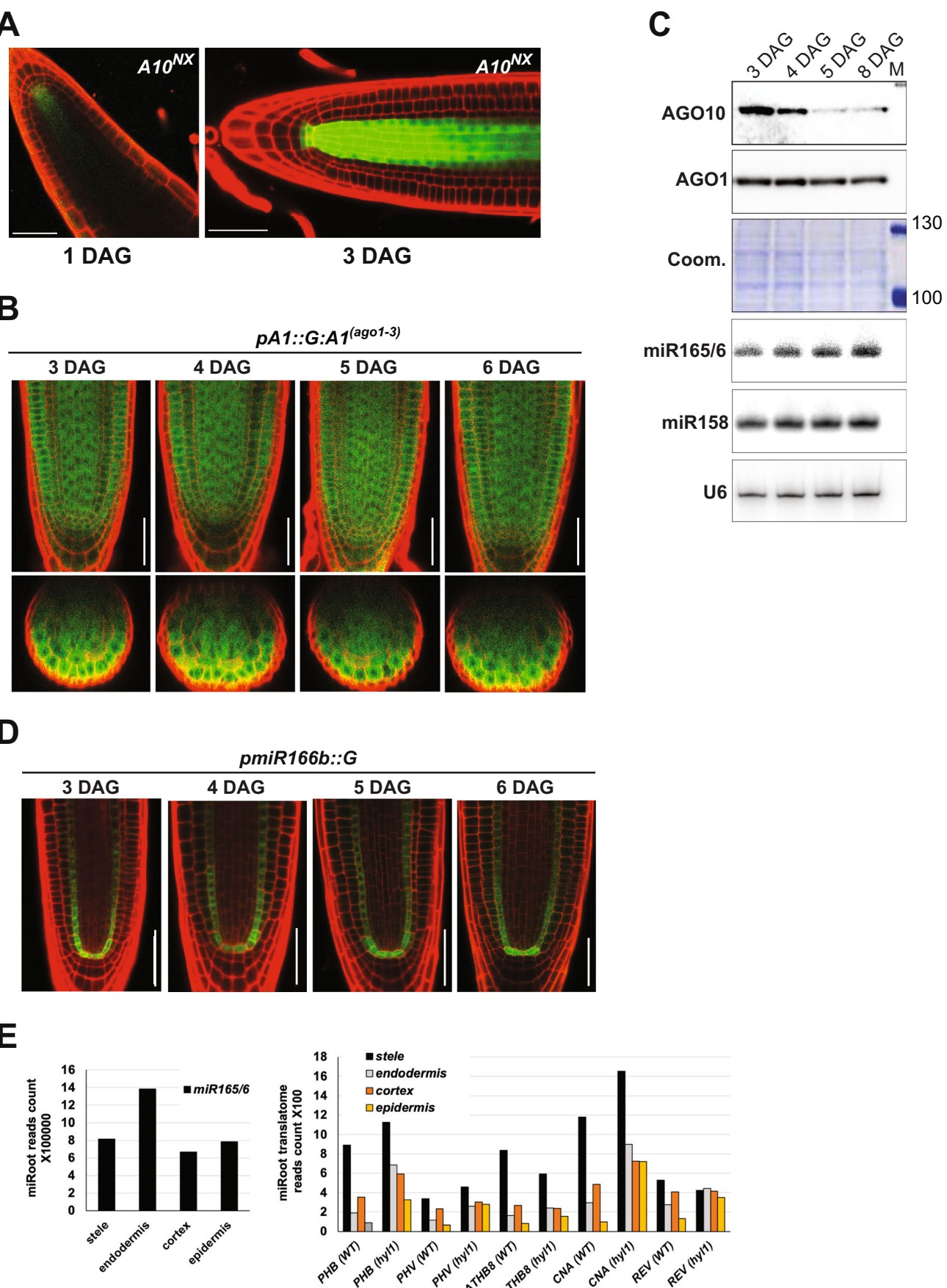

◀  **Figure EV4.   Time-course analyses between 3 and 6 days post-germination.**

(**A**) Compared signal intensity yielded by the *AGO10^NX* reporter at 1-*vs*-3-DAG. (**B**) GFP signal yielded by the *pA1::G:A1^(ago1-3)* reporter. (**C**) Time-course western analysis of endo-AGO1 *vs* endo-AGO10 levels (upper panels) and -northern analysis of miR165/166 vs miR158 levels (lower panels) in whole root tips of WT plants. Coom: Coomassie blue staining provides a total protein loading control. The indicated miRNAs were detected by northern blot analysis via hybridization of radiolabeled sequence-complementary oligonucleotides. Hybridization to the U6 snRNA provides a total small RNA loading control. (**D**) GFP signal yielded by the *pMIR166b::G* transcriptional reporter in the 3-to-6 DAG germination time-course (**E**) miRoot layer-specific reads count for miR165/166 loading into AGO1 (left) and layer-specific translatome analysis of all root-expressed members of the *HD-ZIP III* family in the WT- or miRNA-deficient *hyl1* background (middle). Data information: (**A**) experiment conducted in the main root tip under identical laser settings. (**B**, **D**) seedlings were grown under conditions identical to those used in the 3-to-6 DAG germination time-course of Fig. 6A,C. The same laser settings were applied in acquiring images. Scale bars in (**A**, **B**, **D**): 50 μm. Source data are available online for this figure.

                                                    

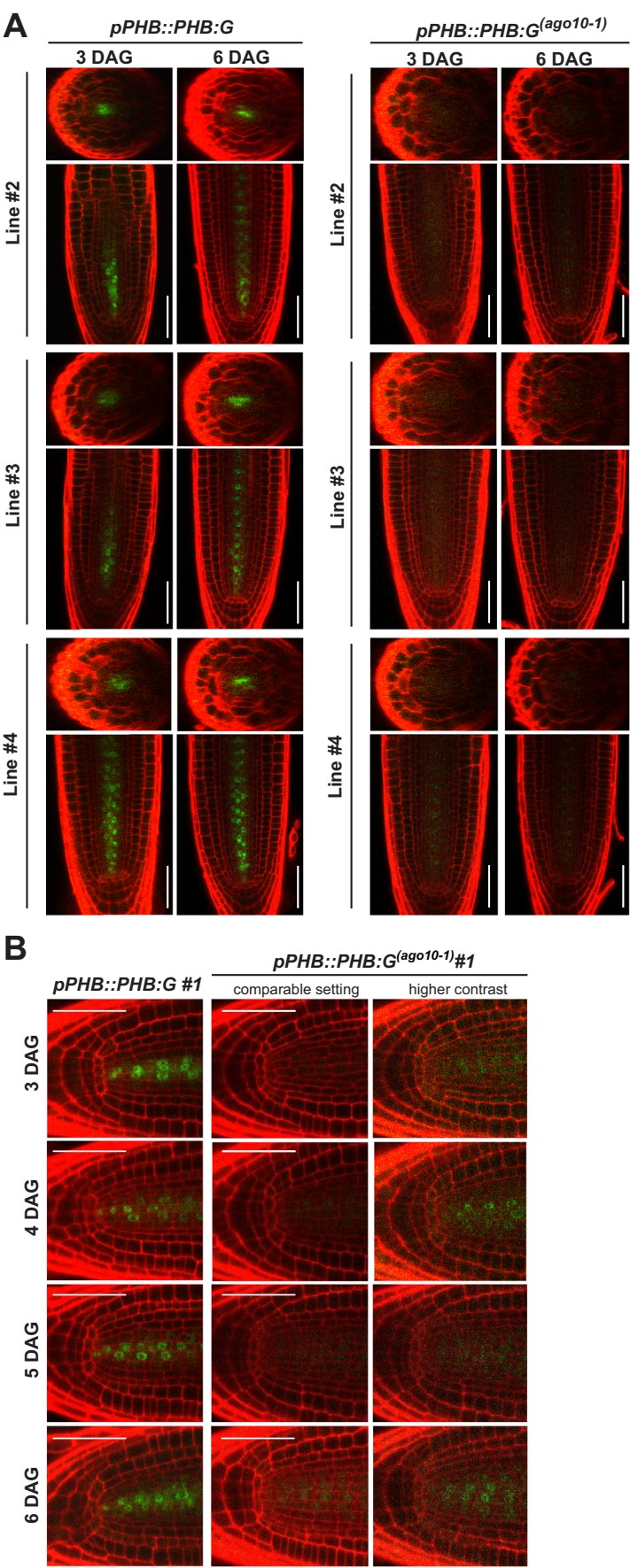

◀ **Figure EV5.   Characterisation of independent lines of pPHB::PHB:G in the WT or *ago10-1* background.**

(A) Longitudinal and radial views of roots of additional independent T2 lines of *pPHB::PHB:G* in either the WT (lines #2, 3 and 4) or *ago10-1* (lines #2, 3 and 4) mutant background at 3 or 6 DAG. Confocal settings were identical to those used in Fig. 6. Scale bars: 50 µm. (B) QC-proximal views of the root tips of lines *pPHB::PHB:G #1* and *pPHB::PHB:G^(ago10-1)* #1 at 3, 4, 5 or 6 DAG. The right panels show *pPHB::PHB:G^(ago10-1)* #1 under normal and enhanced contrast settings to compare to the signal from *pPHB::PHB:G #1*, revealing the wider repartition of PHB in the stele of *ago10-1* background despite substantially reduced levels. Scale bars: 50 µm. Source data are available online for this figure.

