## [Peer Review File · The EMBO Journal]

An AGO10:miR165/6 module regulates meristem activity and xylem development in the Arabidopsis root

Shirin Mirlohi, Grégory Schott, André Imboden, and Olivier Voinnet

Corresponding author(s): Olivier Voinnet (voinneto@ethz.ch)

Review Timeline:

Submission Date:	13th Jan 23
Editorial Decision:	21st Mar 23
Revision Received:	15th Dec 23
Editorial Decision:	12th Feb 24
Revision Received:	21st Feb 24
Accepted:	22nd Feb 24

Editor: William Teale

Transaction Report:

Dear Dr. Voinnet,

Thank you again for the submission of your manuscript entitled "An AGO10:miR165/6 module regulates meristem activity and xylem development in the Arabidopsis root" (EMBOJ-2023-113504) and for your patience during the review process. We have now received the reports from the referees, which I copy below.

As you can see from their comments, the referees state that your work on AGO10 function in the root apical meristem is both important and interesting. That said, Referees 1 and 3 point to significant technical shortcomings which, in their opinion, mean that the conclusions of the manuscript are not supported by the data as they are presented.

However, in my opinion, the requested extra experiments and analyses could be completed in a reasonable period. Therefore, before I invite you to address the comments of all referees in a revised version of the manuscript, I think it would be useful to have a Zoom call to discuss whether the referees' points would be addressable in a single round of revision, or whether it may be more prudent to submit the work elsewhere. It is The EMBO Journal policy to allow only a single major round of revision and that it is therefore important to resolve the main concerns at this stage.

I would also like to point out that as a matter of policy, competing manuscripts published during this period will not be taken into consideration in our assessment of the novelty presented by your study ("scooping" protection). We have extended this 'scooping protection policy' beyond the usual 3 month revision timeline to cover the period required for a full revision to address the essential experimental issues. Please contact me if you see a paper with related content published elsewhere to discuss the appropriate course of action.

Again, please contact me at any time during revision if you need any help or have further questions.

Thank you very much again for the opportunity to consider your work for publication. I look forward to your revision.

Best regards,

William

William Teale, Ph.D.
Editor
The EMBO Journal

When submitting your revised manuscript, please carefully review the instructions below and include the following items:

- 1) a .docx formatted version of the manuscript text (including legends for main figures, EV figures and tables). Please make sure that the changes are highlighted to be clearly visible.
- 2) individual production quality figure files as .eps, .tif, .jpg (one file per figure).
- 3) a .docx formatted letter INCLUDING the reviewers' reports and your detailed point-by-point response to their comments. As part of the EMBO Press transparent editorial process, the point-by-point response is part of the Review Process File (RPF), which will be published alongside your paper.
- 4) a complete author checklist, which you can download from our author guidelines ([https://wol-prod-cdn.literatumonline.com/pb-assets/embo-site/Author Checklist%20-%20EMBO%20J-1561436015657.xlsx](https://wol-prod-cdn.literatumonline.com/pb-assets/embo-site/Author%20Checklist%20-%20EMBO%20J-1561436015657.xlsx)). Please insert information in the checklist that is also reflected in the manuscript. The completed author checklist will also be part of the RPF.
- 5) Please note that all corresponding authors are required to supply an ORCID ID for their name upon submission of a revised manuscript.
- 6) We require a 'Data Availability' section after the Materials and Methods. Before submitting your revision, primary datasets produced in this study need to be deposited in an appropriate public database, and the accession numbers and database listed under 'Data Availability'. Please remember to provide a reviewer password if the datasets are not yet public (see <https://www.embopress.org/page/journal/14602075/authorguide#datadeposition>). If no data deposition in external databases is

needed for this paper, please then state in this section: This study includes no data deposited in external repositories. Note that the Data Availability Section is restricted to new primary data that are part of this study.

Note - All links should resolve to a page where the data can be accessed.

8) For data quantification: please specify the name of the statistical test used to generate error bars and P values, the number (n) of independent experiments (specify technical or biological replicates) underlying each data point and the test used to calculate p-values in each figure legend. The figure legends should contain a basic description of n, P and the test applied. Graphs must include a description of the bars and the error bars (s.d., s.e.m.).

9) We would also encourage you to include the source data for figure panels that show essential data. Numerical data can be provided as individual .xls or .csv files (including a tab describing the data). For 'blots' or microscopy, uncropped images should be submitted (using a zip archive or a single pdf per main figure if multiple images need to be supplied for one panel). Additional information on source data and instruction on how to label the files are available at .

10) We replaced Supplementary Information with Expanded View (EV) Figures and Tables that are collapsible/expandable online (see examples in <https://www.embopress.org/doi/10.15252/embj.201695874>). A maximum of 5 EV Figures can be typeset. EV Figures should be cited as 'Figure EV1, Figure EV2" etc. in the text and their respective legends should be included in the main text after the legends of regular figures.

12) Our journal encourages inclusion of *data citations in the reference list* to directly cite datasets that were re-used and obtained from public databases. Data citations in the article text are distinct from normal bibliographical citations and should directly link to the database records from which the data can be accessed. In the main text, data citations are formatted as follows: "Data ref: Smith et al, 2001" or "Data ref: NCBI Sequence Read Archive PRJNA342805, 2017". In the Reference list, data citations must be labeled with "[DATASET]". A data reference must provide the database name, accession number/identifiers and a resolvable link to the landing page from which the data can be accessed at the end of the reference. Further instructions are available at .

Additional instructions for preparing your revised manuscript:

At EMBO Press we ask authors to provide source data for the main manuscript figures. Our source data coordinator will contact you to discuss which figure panels we would need source data for and will also provide you with helpful tips on how to upload

and organize the files.

We realize that it is difficult to revise to a specific deadline. In the interest of protecting the conceptual advance provided by the work, we recommend a revision within 3 months (19th Jun 2023). Please discuss the revision progress ahead of this time with the editor if you require more time to complete the revisions. Use the link below to submit your revision:

Referee #1:

In this manuscript the authors are studying the role of ARGONAUTE10 in the root vasculature formation. They suggest that AGO10 quenches miR165/6 allowing the silencing of HD-ZIP III by AGO1. This ends in controlling cell division and xylem differentiation. The authors use genetic approach to indicate that AGO10 regulates the RAM length/activity. In addition, the authors suggest that AGO10 is needed for stele development modulating the formation of proto and metaxylem. Besides, they study the cell type and subcellular localization of AGO10-AGO1 interaction in relation with the localization of the miR165/6. This is an interesting and original study combining cell biological and developmental aspects of the miR165/6-HD ZIP III regulon. However, at this stage there are several complications that have to be resolved (see below). Also, as a general comment, this manuscript is difficult to read and to understand. The sentences are too long and encrypted.

Major comments:

The authors claim that the mutant ago10-1 has larger meristem size than the WT but with the n used I doubt this difference is significant, especially after comparing images 2A with 2C. It is known that trichoblasts and atrichoblasts display different length of the cell cycle. Shorter cell cycle of trichoblasts makes cells shorter, hence, there is more of them in the individual epidermal cell file. Depending on the position of the optical cross section, the confocal image may show trichoblasts, atrichoblasts or both. In figure 2C, ago10-1 image is showing different number of cells on the left than on the right side. The authors quantified the number of cells on the right half of the image (30 cells) while number of epidermal cells, in the very same image, on the left side shows 19 cells, what corresponds to Wt phenotype. This may lead to the bias in the analysis e.g. in the image of ago10-1 (Fig 2C). The authors should explain the reasons why they selected this side for quantification and not the other. Assuming the image of ago10-1 in Fig 2C is representative to the whole data set, the lack of 19 cell data point in the Fig2 D, clearly shows biased analysis. In such case I would request removal of quantification of epidermal cells. Along the same line, following most of the literature, I would also suggest using length of cortex cells for estimation of the root meristem length (Fig 2A). Related to this; Figure 2F. Could the authors indicate in which area are these pictures done? The authors should also make sure that similar error is not introduced to the quantification of vascular cells, where different cell types show different morphology.

The authors are analyzing xylem specification close to the differentiation zone of the root by Fuchsin staining. The phenotype they show for ago10 mutant is not similar to the one showed in El Arbi, which appears to make a conflict that should be rectified prior to the prospective publication. They indicate that their observation are done in an earlier stage but according to the information in both papers, both experiments are done in 6 days old plants. Besides, the authors say "Conversely, Fuchsin staining was more QC-proximal in A10OX- than in WT- roots, presumably reflecting A10OX reduced RAM length/activity (Figures 2A-C; S3) and/or premature differentiation. 17-out-of-20 inspected A10OX primary roots either lacked PX cells altogether (MX-only phenotype likely due to reduced periclinal divisions, Figure 2E-F) and/or displayed one or two additional (on top of the two cognate) MX cells (Figure 4F-G)." These can be not know with the pictures they are showing in figure 2A and C because they do not use Fuchsin staining in these and, in figure 4 we can not know if the area is close to the RAM.

It is not clear why the authors analyze the sdn mutant phenotype but not the HD-ZIP III mutants. Specially because the authors propose a new role ("pincer") for AGO10 protein in regulation of miR165/6 levels in the root meristem. The main target of miR165/6 in the RAM are class III HD-ZIPs. In order to support their claims and models the authors should report HD-ZIP (PHB, CNA) protein accumulation patterns in ago10 genetic background in studied cell types (vasculature, initials) and at studied developmental time points (1-6 days after germination).

Is figure 5D mentioned in the text?

The image of the root meristem cross-section with pPHB::PHB-GFP in Fig 6D does not match expression pattern found in cited, previously published work (Miyashima et al., 2011 Fig4B show relatively broad distribution of PHB in the vascular cells; Carlsbecker et al., 2010 Fig 2a shows accumulation of PHB mRNA in metaxylem and procambial cells). It looks like very high contrast was applied what would diminish lower level signal in the procambial cells. In fact cited literature shows gradient of HD-ZIP mRNA with highest levels in the metaxylem cells. Can authors provide an image with enough gain and as little contrast so that the noise just starts to appear in the background of the image? Can authors provide longitudinal section of the meristem to assess protein accumulation along the meristem? If PHB/CNA accumulates in a gradient fashion, the authors may want to use "fire"-type LUT for the green channel.

The authors conclude: "Therefore, GFP:AGO10 displays both high affinity and selectivity for miR165/166 in the RAM, supporting a model whereby an AGO10-vs-AGO1 competition for miR165/166 modulates the extent of PHB silencing including, possibly, in root layers' initials (Figure 2G)." but did not provide any data supporting cell-division limiting role of AGO10/HD-ZIP III in the root initials. The authors should substantiate this statement by providing evidence based on live-imaging or EdU staining or based on some other quantitative and informative method. Alternatively, the discussion on the potential role of the AGO10/HD-ZIP III regulation in root elongation could be significantly downplayed.

It is not clear why they can not detect AGO10 in Figure 3A by using GFP antibody if they detected it in figure 1B. However, they detect the miRNAs in AGO10 sample and the authors conclude saying that "AGO10 displays substantially higher affinity for these cargoes than AGO1". Why there is no sample loading detection in the coomassie gel for the IPs in figure 3A and figure 5C? With the amount of GFP detection in A1 sample, we should see some band in the Coomassie gel. In addition, in line 216 is wrong they are referring to the western blot in Figure 1A but it should be 1B.

Minor:

Several typos in the main text should be corrected.

Referee #2:

The authors take on a very important challenge to uncover the role of AGO10 in regulating the RAM length/activity, root xylem specification by quenching the miR165/166 ability to silence HD-ZIP III transcripts. Although I think the authors make here a very important contribution, I have some comments concerning certain statements:

- I think the reference to El Arbi et al., 2021 would be appropriate here too, as the parallel study also shows the transcriptional activity of AGO10.
- Figure 2A. The arrow in shr-2 mutant is not pointing at elongation onset and the next cell is clearly not twice as size as previous ones, please indicate the appropriate spot.
- Figure 4. In The Basic Fuchsin stained images it would be very informative if the authors indicate which stage of xylem formation is shown - onset, particular distance from the root tip or other developmental indicator. Is the phenotype shown in Fig.4F persistent throughout the root? In case where one or more xylem files are missing, do they appear with a delay later during development?
- Line 351, Figure 5D and S3B. I think that the treatment with leptomycin B should also include a control excluding that the cells are still viable after LMB treatment prior clearing and processing for cell wall staining. A simple test would be to stain the treated and mock roots with propidium iodide which will stain the nuclei of dead cells.
- Figure 2G model. How are HD-ZIP III behaving in both ago10 and AGO10 overexpression lines? I think it would be very informative to show the behavior of existing tagged lines of HD-ZIP IIIs to support the model presented. The phenotypes of AGO10 mutants and overexpression lines are very convincing and indicating towards the link with HD-ZIP IIIs. Still, the behavior of HD-ZIP IIIs should be analyzed in corresponding backgrounds.
- Figure 4H. As most of roots of sdn1sdn2 show 2px and no mx, is this phenotype consistent throughout the root or perhaps the formation of metaxylem is delayed?

Referee #3:

Mirlohi et al report on the roles of AGO1 and AGO10 in confining HD-ZIP III transcription patterns in the Arabidopsis root through miR165/6. There are several unanswered questions in this area that the study approaches. While there are some interesting

observations, the study is somewhat incomplete and inconclusive. There are several points that need to be extended upon or clarified, listed below:

Firstly, the writing style is convoluted and elaborate. Many words are used to describe relatively simple and obvious observations and their implications. I would encourage the authors to streamline the text. Where possible, the use of less convoluted, shorter sentences would help the reader.

Specific points:

1. The conclusions related to AGO10-GFP expression levels relative to WT levels are problematic without proper quantification. It is clear that there is a gradient of accumulation levels among the lines used, but none of these seems to have WT levels.
2. It is strange to conclude that AGO10 "prevails" in the RAM without actually comparing AGO10 and AGO1 levels in these cells.
3. The conclusion that post-transcriptional control determines the proximal-to-distal accumulation gradient of AGO10 in the root stele is not trivial to make. To ensure that the gradients are indeed different, the authors would have to quantify levels of GFP fluorescence at multiple defined positions relative to the QC and determine gradient steepness. An alternative interpretation of their data is that the AGO10-GFP protein is simply less stable than free GFP, as is the case for many other proteins.
4. The depiction of y-axes in graphs (e.g. in Figures 2B,D,E) exaggerates differences, as the origin is not 0. I would propose to clearly indicate that the axis is interrupted, and a segment is shown.
5. When comparing AGO1 and AGO1 in their binding (affinity) to miR165/6, the strategy used is not really giving an answer. The authors do correct for the narrow versus broad expression pattern, but this does not do justice to differences that may exist within the relevant cell type. The proper experiment would have been to express AGO1-GFP from the AGO10 promoter and compare lines directly. As presented, the data is consistent but does not prove, a higher affinity of AGO10 to miR165/6, compared to AGO1.
6. The experiments with P19 are consistent with the model the authors propose, but it is not entirely clear which point the authors wish to make. Movement of miR165/6 has been demonstrated previously, so mopping these miR's up in the endodermis would certainly lead to loss of their role in the stele. Maybe I am missing something important here...
7. The observations on AGO10 pattern evolution over time in the stele are very interesting, and support the notion that AGO10 patterns contribute to the observed HD-ZIP III patterns. A direct test of this role of AGO10 patterns would be to express an additional copy of AGO10 from a xylem-specific promoter. According to the model, this should directly impact HD-ZIP II patterns in a predictable manner.

Referee #1:

In this manuscript the authors are studying the role of ARGONAUTE10 in the root vasculature formation. They suggest that AGO10 quenches miR165/6 allowing the silencing of HD-ZIP III by AGO1. This ends in controlling cell division and xylem differentiation. The authors use genetic approach to indicate that AGO10 regulates the RAM length/activity. In addition, the authors suggest that AGO10 is needed for stele development modulating the formation of proto and metaxylem. Besides, they study the cell type and subcellular localization of AGO10-AGO1 interaction in relation with the localization of the miR165/6.

This is an interesting and original study combining cell biological and developmental aspects of the miR165/6-HD ZIP III regulon.

We thank the referee for their appreciation of the quality and originality of our work. We also thank the referee for the very constructive suggestions of additional experiments or for recommending toning down some aspects, which we followed meticulously. The new experiments have provided crucial additional insights not available in the previous version. We believe these new insights now consolidate the study and its conclusions.

However, at this stage there are several complications that have to be resolved (see below). Also, as a general comment, this manuscript is difficult to read and to understand. The sentences are too long and encrypted.

We have made our best efforts to make the manuscript easier to read by splitting long sentences and using different, more straightforward wording.

Major comments:

The authors claim that the mutant *ago10-1* has larger meristem size than the WT but with the *n* used I doubt this difference is significant, especially after comparing images 2A with 2C. It is known that trichoblasts and atrichoblasts display different length of the cell cycle. Shorter cell cycle of trichoblasts makes cells shorter, hence, there is more of them in the individual epidermal cell file. Depending on the position of the optical cross section, the confocal image may show trichoblasts, atrichoblasts or both. In figure 2C, *ago10-1* image is showing different number of cells on the left than on the right side. The authors quantified the number of cells on the right half of the image (30 cells) while number of epidermal cells, in the very same image, on the left side shows 19 cells, what corresponds to Wt phenotype. This may lead to the bias in the analysis e.g. in the image of *ago10-1* (Fig 2C). The authors should explain the reasons why they selected this side for quantification and not the other. Assuming the image of *ago10-1* in Fig 2C is representative to the whole data set, the lack of 19 cell data point in the Fig2 D, clearly shows biased analysis.

- 1) ***In such case I would request removal of quantification of epidermal cells.***

*The referee is perfectly correct in their assessment that the epidermis should not be used for cell number counting because of the trichoblasts/atrichoblast situation. As we were unaware of this added complexity, we have abandoned our original focus on the epidermis in this section, as recommended. We also have discarded the old data and recounted cell numbers in *n*=30 (instead of 10, previously) independent roots for each indicated genotype.*

- 2) ***Along the same line, following most of the literature, I would also suggest using length of cortex cells for estimation of the root meristem length (Fig 2A).***

We acquired $n=30$ completely new pictures for each genotype, using the cortex, endodermis and the stele for the new counting of longitudinal cell numbers. We also now use the first elongating cortex cell to report meristem length, as calculated from the QC, in the revised Figure 2A. The data in each layer are consistent with *ago10-1* having a longer RAM and *AGO10^{OX}* a shorter RAM, as compared to WT.

- 3) **Related to this; Figure 2F. Could the authors indicate in which area are these pictures done?**

We have now re-done our analysis completely anew with $n=30$ pictures now acquired at two positions in the stele (20 and 120 μm from the QC), as indicated in Fig 2E-F.

- 4) **The authors should also make sure that similar error is not introduced to the quantification of vascular cells, where different cell types show different morphology.**

Although there was no overt error in our first analysis of the stele, we have been very careful to count the cells within the stele of new roots ($n=30$ new pictures for each genotype) at the indicated two positions in the pictures now shown in Figure 2E-F. At both QC-proximal and QC-distal regions, the stele contains statistically more versus less cells in *ago10-1* versus *AGO10^{OX}*, respectively, compared to WT. These results are in line with those from El Arbi et al. (2023).

5) The authors are analysing xylem specification close to the differentiation zone of the root by Fuchsin staining. The phenotype they show for *ago10* mutant is not similar to the one showed in El Arbi, which appears to make a conflict that should be rectified prior to the prospective publication. **They indicate that their observation are done in an earlier stage but according to the information in both papers, both experiments are done in 6 days old plants.** Besides, the authors say "Conversely, Fuchsin staining was more QC-proximal in *A10OX-* than in *WT-* roots, presumably reflecting *A10OX* reduced RAM length/activity (Figures 2A-C; S3) and/or premature differentiation. 17-out-of-20 inspected *A10OX* primary roots either lacked PX cells altogether (MX-only phenotype likely due to reduced periclinal divisions, Figure 2E-F) and/or displayed one or two additional (on top of the two cognate) MX cells (Figure 4F-G)." These can be not know with the pictures they are showing in figure 2A and C because they do not use Fuchsin staining in these and, **in figure 4 we can not know if the area is close to the RAM.**

We are very sorry for the lack of clarity in our statement. By "earlier stage" we did not mean "days after germination", but "developmental time over the longitudinal axis" where cells age as they are pushed away from the QC, elongate and differentiate. In other words, we intended to point out that our observations were QC-proximal whereas those of El Arbi were QC-distal i.e. beyond the differentiation zone. We have now clarified the text and solved this issue by imaging xylem specification at two positions, QC-proximal (as in the first version) and QC-distal. Our QC-distal observations are completely in line with those of El Arbi et al., 2023 (theirs were conducted in the QC-distal zone only), including observations of their *sgo1* allele ($n= 22$ roots) under our growth conditions, as now shown in Figure EV3E. Note that Figure 4D was added to help the reader locate the respective positions of the two sections relative to the *AGO10* expression domain, using line *A10^{NX}*. We have, finally, better documented the Fuchsin staining starting-points in WT, *ago10-1* and *A10^{OX}* roots in Figure EV3C.

We hope these clarifications on wording and new analyses conducted at two positions with many more roots now address the referee's legit issues. We have also included

supplementary file 3 to document precisely the PX versus MX cell number for each individual plant of each genotype as used to produce Figure 4. These data were collected early after cell division (i.e. QC-proximally; QP) and late after cell division (i.e. QC-distally; QD).

6) it is not clear why the authors analyze the sdn mutant phenotype but not the HD-ZIP III mutants.

As stated in the introduction and further in the main text, the action of AGO10 in the SAM is not only to selectively bind miR165/166 competitively to AGO1, but also to promote miR165/166 turnover via the activities of AGO10-coupled nucleases called SDN1 and SDN2. In other words, the turnover of AGO10-bound miR165/6 depends on the SDNs' function in the shoot. New immunoprecipitation data added in Figure 3B now show that, as previously showed in the SAM, more miR165/166 is loaded into AGO1 in the ago10-1 compared to WT background in the RAM. This is unlike the abundant miR158, for which AGO10 has no specific affinity, hence its equal loading in AGO1 in both backgrounds. This new result agrees with the notion that the AGO10-bound miR165/166 pool is degraded in the RAM, with SDN1/2 being likely involved.

The likely involvement of SDN1/2 made it relevant, therefore, to inspect the xylem differentiation phenotype of sdn1 sdn2, because it was expected to at least partly resemble that of ago10-1. This is indeed what we found because these roots lack MX altogether.

We had not tested higher-order HD ZIP III mutant roots (expected also to resemble ago10-1) because El Arbi et al. in their 2022 version had already tested the phb phv can triple mutant in their own manuscript. However, they kindly provided us seeds to test them under our own growth conditions, which we did in both QC-proximal and QC-distal regions of the roots. The results, now presented on Figure 4F, are fully consistent with an ago10-1-like phenotype (which should lead to enhanced HD-ZIP III silencing) in both QC-proximal and QC-distal regions.

7) Specially because the authors propose a new role ("pincer") for AGO10 protein in regulation of miR165/6 levels in the root meristem. The main target of miR165/6 in the RAM are class III HD-ZIPs. In order to support their claims and models the authors should report HD-ZIP (PHB, CNA) protein accumulation patterns in ago10 genetic background in studied cell types (vasculature, initials) and at studied developmental time points (1-6 days after germination).

We recognize that the analysis of HD ZIP III in the WT versus ago10-1 background was missing in our original manuscript. It is indeed a corner stone of both our and El Arbi et al.'s manuscript.

To address the valid referee's point given our inability to conduct in situ hybridizations, we decided to focus on the PHB:GFP fusion, since PHB was previously identified as a key determinant to xylem differentiation among the HD-ZIP IIIs. Upon careful inspection, we realised that the pPHB::PHB:GFP line obtained from the Benfey lab had retained very low GFP expression levels, if any, possibly due to co-suppression, as shown below:

We thus resorted to engineer new transgenic lines using the pPHB::PHB:GFP construct from Miyashima et al., 2011 kindly sent to us by Dr Miyashima. The obtention and characterization of multiple independent lines (WT versus ago10-1 background) is explained in the material and methods of the revised manuscript.

We performed all the analyses requested by the referee in both genetic backgrounds, both in the initials and vasculature at 6 DAG (Figure 2H) as well as dynamically, during early germination from 3 to 6 DAG (Figures 6, EV6A-B). We also systematically used the pPHB::H2B:G transcriptional reporter in parallel to document the PHB expression domain in the stele and initials. This, we believe, now convincingly shows negative post-transcriptional regulation by miR165/166 modulated in an AGO10-dependent manner. Essentially, we lose accumulation of PHB::GFP in the stele and its initials in ago10-1, consistent with a protective role for AGO10, as predicted by our model.

See also our answers to points 9-10 for additional, key aspects we discovered in the course of this new analysis, which, we think, also bolster our model (Figure 7).

8) Is figure 5D mentioned in the text?

We thank the referee for spotting this error. It is now referenced where appropriate. We verified to the best of our abilities that no other figure panel has been left unreferenced.

9) The image of the root meristem cross-section with pPHB::PHB-GFP in Fig 6D **does not match expression pattern found in cited, previously published work** (Miyashima et al., 2011 Fig4B show relatively broad distribution of PHB in the vascular cells; Carlsbecker et al., 2010 Fig 2a shows accumulation of PHB mRNA in metaxylem and procambial cells). It looks like very high contrast was applied what would diminish lower level signal in the procambial cells. In fact cited literature shows gradient of HD-ZIP mRNA with highest levels in the metaxylem cells. **Can authors provide an image with enough gain and as little contrast so that the noise just starts to appear in the background of the image?**

We agree with the referee about a possible discrepancy with previous work, which we had indeed noted in our first version, leading us to ask how a radiating gradient might become focussed onto the xylem axis? We note, however, that our observations –made at later stages of germination (where the GFP signal is centred along the xylem axis and appears almost restricted to the MX)– are in agreement with more recently published PHB in situ hybridization data from Fan et al., 2021 (<https://doi.org/10.1073/pnas.2022547118>):

In fact, the request of the referee to analyse HD-ZIP III in WT versus ago10-1 led us to discover that the PHB:GFP transcripts levels and the shape of the gradient evolve over time during early germination. Our new analysis shows that they strikingly mirror the levels and shape of GFP:AGO10 and endoAGO10 (Figure 4H, Figure EV5A-D, Figure 6 and Figure EV6A-B).

We found that, at early stages (3-4 DAG), GFP:AGO10 and endoAGO10 accumulate to high levels inside the stele, though less so in the pericycle; GFP:AGO10 is not yet depleted from the xylem axis. PHB is lowly transcribed at these early time points and thus protected by abundant AGO10 in most of the stele. The miR165/166 gradient is thereby radial, with maximal silencing on the outer part (e.g. pericycle, containing less AGO10) and minimal silencing inside it. The ensuing picture, both longitudinally and radially, is essentially the same as reported by Miyashima et al., 2011 (Figure 4B) using the same pPHB::PHB:GFP reporter. So, there is no longer a discrepancy here.

However at later time points (5-8 DAG), GFP:AGO10 and endoAGO10 levels drop substantially and GFP:AGO10 progressively acquires the pincer shape. Transcript levels of PHB:GFP (as inferred from the pPHB::H2B:GFP reporter) increase strongly over the same time window. This likely explains why the intensity (as opposed to shape, which is progressively focussed along the xylem axis) of the gradient's signal remains approximately the same all along the time course. The gain in PHB transcripts throughout the stele (and beyond) now presumably provides a means to consume the abundant mobile miR165/166 via AGO1-dependent target-directed miRNA degradation (TDMD; Shi et al. 2020; 10.1126/science.abc9359). This was indeed already anticipated by Carlsbecker et al. in 2010.

We infer that the remaining miR165/166 pool not consumed by TDMD would then be buffered by AGO10, but only within AGO10's accumulation domain (pincer), leaving a presumably small miR165/166 fraction available for gradient formation along the AGO10-free xylem axis. This possibly explains the now more linear shape of the HD-ZIP III gradient at 6 DAG and beyond. Our time course analysis shows that at about 8 DAG the signal is essentially focussed in the 2/3 MX cells inside the axis, as we had already reported in the first version of the study. This pattern, which does not evolve further beyond 8 DAG, is in line with the recent *in situ* hybridization patterns of Fan et al. 2021 shown above. So, there is no discrepancy here either.

Thus, we do not think that the axis-focussed and MX-centred pattern seen at later germination stages were caused by high contrast applied to the original images. In fact, we made absolutely sure that identical laser settings were used all throughout the new time course analysis presented in Figure 6AC-D and Figures EV5AC-D, EV6) and involving all the contenders, with minimal or no post-processing of the images, including the contrast. Instead, the time course reveals a highly dynamic process whereby the PHB gradient is being spatially refined with the help of AGO10 over a narrow yet likely decisive time frame. AGO10 is key to the process because in its absence: (i) the PHB:GFP signal is very low and (ii) what remains of it is more widespread throughout the stele, as opposed to focussed along the axis in WT (Figure 6A, 6C, EV6B).

From these new analyses, we propose that the PHB patterns reported by Miyashima et al., 2011 (and to a lesser extent Carlsbecker et al. 2010) and by Fan et al. 2021 are both correct but they were likely imaged at different time points post-germination. In fact none of these studies indicates precisely their sampling times, which was not stringently required at the time because the highly dynamic process described here (over just a few days) was simply unknown. We now allude to these differences in the revised discussion to point out how our new observations with AGO10 now reconcile the previous findings of our colleagues, all of which were entirely correct, albeit made, most likely, at slightly distinct time points.

10) Can authors provide longitudinal section of the meristem to assess protein accumulation along the meristem? If PHB/CNA accumulates in a gradient fashion, the authors may want to use "fire"-type LUT for the green channel.

At the request of another referee, we have now confirmed, in Figure 1C, that GFP:AGO10 forms a QC-proximal-to-QC-distal gradient along the stele's longitudinal axis at 6 DAG. A new set of analyses presented in Figure 6E (6 DAG), also shows that the PHB:GFP longitudinal signal is similarly graded and its end-point (marked by a dashed line) strikingly

coincides with that of the GFP:AGO10 signal. By contrast, both the PHB and cumulated miR165/166 transcriptional reporters continue yielding strong signals beyond this zone. This further supports the notion that mobile miR165/166 post-transcriptionally silences PHB in a manner buffered by the AGO10 pincer. Note that the upper section of Figure 6E#5 is also consistent with the highly reduced PHB::GFP accumulation seen by us in the ago10-1 background.

11) The authors conclude: "Therefore, GFP:AGO10 displays both high affinity and selectivity for miR165/166 in the RAM, supporting a model whereby an AGO10-vs-AGO1 competition for miR165/166 modulates the extent of PHB silencing including, possibly, in root layers' initials (Figure 2G)." **but did not provide any data supporting cell-division limiting role of AGO10/HD-ZIP III in the root initials. The authors should substantiate this statement by providing evidence based on live-imaging or EdU staining or based on some other quantitative and informative method.**

Alternatively, the discussion on the potential role of the AGO10/HD-ZIP III regulation in root elongation could be significantly downplayed.

We believe that the key addition, in Figure 2H, of the pPHB::2B:G transcriptional reporter and pPHB::PHB:G translational reporter (in both WT and ago10-1 in this case) makes a better case for our model. It also shows that PHB is indeed expressed in the epidermis and ELRCI, but at lower levels than in the other layers. Thus, the substantial and mostly speculative part pertaining to the epidermis initially present in the manuscript has been entirely removed (incidentally fulfilling a request of the referee). Thus, this whole section now focuses solely on our new findings.

EdU staining was used by El Arbi et al. 2023 to show that cell division is impacted in the stele of sgo1 and ago10-1. However, we agree with the referee that this has not been assessed in the initials of the ground tissue, for instance. We now specifically state this at the end of the corresponding section to downplay the claims and indicate that more work will be needed to fully understand the role of AGO10 on RAM length/activity control. We have accordingly carefully removed the words 'division' and 'differentiation' in this whole section and replaced them by 'meristem length/activity' where applicable. Finally, we have also changed the model in Figure 2G to now simply refer to 'short RAM' or 'long RAM' instead of 'division' and 'differentiation', to take into account the legit comment of the referee.

12) It is not clear why they cannot detect AGO10 in Figure 3A by using GFP antibody if they detected it in figure 1B. However, they detect the miRNAs in AGO10 sample and the authors conclude saying that "AGO10 displays substantially higher affinity for these cargoes than AGO1".

The reviewer compares the results of Figure 1B with those of Figure 3A, yet these figures report very different experiments: either a standard western (1B) or an immuno-precipitation (IP; 3A). These rely on distinct extraction procedures, as detailed in the material and methods. Western experiments are conducted with the Tanaka extraction procedure in which phenol extraction followed by precipitation yields an optimal amount of total protein concentrated in a small volume of 50 µL. IP experiments involve native extractions via a lysis buffer without any precipitation, and resuspension in 1 mL of which 100 µL (1/10) serve as the basis for the input and IP samples, of which 20 µL are then loaded. Thus, the amounts of proteins involved in the respective experiments are incomparable.

Why there is no sample loading detection in the coomassie gel for the IPs in figure 3A and figure 5C? With the amount of GFP detection in A1 sample, we should see some band in the Coomassie gel.

See our answer above: the left-hand sides of the gels are westerns, whereas the right-hand sides are IPs, with correspondingly very dissimilar amounts of inputs. It is, in fact, very difficult to detect AGO1 (or any AGO for that matter) upon Coomassie staining of an IP as many publications in the plant field attest to. Here is another example from our lab, where 4 IPs were conducted with anti-AGO1 on 2 X (WT versus mutant genotypes). Despite a massive (>20 folds) enrichment of the protein after IP, AGO1 is below Coomassie staining detection (yellow rectangles).

13) In addition, in line 216 is wrong they are referring to the western blot in Figure 1A but it should be 1B.

Thank you for spotting this mistake, which has now been corrected.

Minor:

Several typos in the main text should be corrected.

We have tried our best to detect and to correct all typos by carefully re-reading and re-writing the manuscript.

Supporting literature for Referee #1

Fan P, et al. The receptor-like kinases BAM1 and BAM2 are required for root xylem patterning. *Proc Natl Acad Sci U S A*. 2021 Mar 23;118(12):e2022547118. doi: 10.1073/pnas.2022547118.

Shi CY, et al. The ZSWIM8 ubiquitin ligase mediates target-directed microRNA degradation. *Science*. 2020 Dec 18;370(6523):eabc9359. doi: 10.1126/science.abc9359.

Referee #2:

The authors take on a very important challenge to uncover the role of AGO10 in regulating the RAM length/activity, root xylem specification by quenching the miR165/166 ability to silence HD-ZIP III transcripts. Although I think the authors make here a very important contribution, I have some comments concerning certain statements:

We thank the referee for their appreciation of our work and for making constructive suggestions, which we have strived to follow to improve our study and manuscript.

1) I think the **reference to El Arbi et al., 2021** would be appropriate here too, as the parallel study also shows the transcriptional activity of AGO10.

We had referenced the original manuscript reporting the NT transcriptional fusion used by El Arbi in their 2021 preprint. Since El Arbi et al.'s work has vastly evolved since nearly 3 years, we believe it is probably better to cite their accompanying manuscript instead, which we do extensively.

2) Figure 2A. The arrow in shr-2 mutant is not pointing at elongation onset and the next cell is clearly not twice as size as previous ones, **please indicate the appropriate spot.**

We have now positioned the onset of the elongation zone using the cortex (instead of epidermis) as a reference and believe the arrows are now set correctly.

3) Figure 4. In The Basic Fuchsin stained images it would be very informative if **the authors indicate which stage of xylem formation is shown** - onset, particular distance from the root tip or other developmental indicator.

This is indeed a very valid suggestion, which was also made by Referee #1 in relation to the fact that our initial observations were made QC-proximally, whereas those of El Arbi et al. were made QC-distally. We have now used both QC-proximal and -distal observations in all our new xylem studies in the amended Figure 4 to dispel any apparent discrepancy between the two studies. Indeed, the new QC-distal observations are now completely in line with those of our colleagues. We also added the picture in Figure 4D to indicate the two positions studied using the distance from the QC as a baseline.

Is the phenotype shown in Fig.4F persistent throughout the root? In case where one or more xylem files are missing, do they appear with a delay later during development?

See also our answer above. Since we have now imaged both QC-proximally and -distally, it is apparent that MX files missing in the QC-proximal zone eventually develop in the QC-distal zone in many individuals with the ago10-1 (Figure 4E) and phb phv can^(er-2) triple mutant backgrounds (the latter was added upon request by referee #1; Figure 4F). This delayed (as opposed to absent) MX development is now mentioned in the main text. Note, additionally, that due to higher cell numbers in the stele of ago10-1 and phb phv can^(er-2), the number of initially delayed MX cells actually increases QC-distally as compared to WT. Finally, the absent PX in the QC-proximal zone of AGO10^{OX} is partially recovered QC-distally and only in a fraction of individuals (Figure 4H).

4) **Line 351, Figure 5D and S3B. I think that the treatment with leptomycin B should**

also include a control excluding that the cells are still viable after LMB treatment prior clearing and processing for cell wall staining. A simple test would be to stain the treated and mock roots with propidium iodide which will stain the nuclei of dead cells.

We are sorry that we are not entirely sure about the referee's point here. Do they mean, perhaps, that the cause for nuclear signals after LMB treatments is cell death? Transient LMB treatment as performed in our study is an accepted standard in protein nucleo-cytoplasmic shuttling studies in plants. It was conducted essentially as in Bologna et al. 2018, in which AGO1 was shown to shuttle between the nucleus and the cytosol. The only difference is that a vacuum was applied to allow LMB to penetrate the inner root and the stele, where AGO10 resides preponderantly. There really is no specific reason to believe that the vacuum would cause cell death to an extent that would explain the nuclear pattern seen only with the LMB-coupled but not mock-coupled vacuum treatment. Finally, propidium iodide staining is incompatible with the procedure as we perform it.

*Nonetheless, to bolster the proposal that, like AGO1, AGO10 is nucleo-cytoplasmic, we have performed new experiments based on transient expression in *N.benthamiana* leaves, which we had also employed in Bologna et al. 2018 using the GFP:AGO1 construct. We compared the subcellular distribution of free GFP (used as a positive control, since it diffuses in the nucleus from the cytosol), GFP:AGO1 and GFP:AGO10. The new results (now shown in Figure 5F) indicate that GFP:AGO10, just like GFP:AGO1, is nucleo-cytoplasmic. We are hopeful that this new experiment now addresses the referee's point.*

5) Figure 2G model. How are HD-ZIP III behaving in both ago10 and AGO10 overexpression lines? I think it would be very informative to show the behaviour of existing tagged lines of HD-ZIP IIIs to support the model presented. The phenotypes of AGO10 mutants and overexpression lines are very convincing and indicating towards the link with HD-ZIP IIIs. **Still, the behaviour of HD-ZIP IIIs should be analysed in corresponding backgrounds.**

This is a perfectly valid request from the referee, which was also made by one of the other reviewers as a necessary step to bolster our model.. It is indeed a corner stone of both our and El Arbi et al.'s manuscript.

Since PHB is, among the HDZIP IIIs, crucial to xylem differentiation, we focused on pPHB::PHB::GFP. We first analysed the line employed by the Benfey lab in the seminal Carlsbecker et al. (2010) paper but found that the GFP signal was too low to be exploitable, possibly due to co-suppression:

We thus resorted to engineer new transgenic lines using the pPHB::PHB::GFP construct from Miyashima et al., 2011 kindly sent to us by Dr Miyashima. The obtention and characterization of multiple independent lines (WT versus ago10-1 background) is explained in the material and methods of the amended manuscript.

We performed the analyses requested by the referee in both genetic backgrounds, both in the initials and vasculature at 6 DAG (Figure 2H) and dynamically during early germination (Figures 6, EV6A-B). We also systematically used the pPHB::H2B::GFP transcriptional reporter in parallel to document the PHB expression domain in the stele and initials, and highlight negative post-transcriptional regulation by miR165/166 modulated in an AGO10-dependent manner. Essentially, we lose accumulation of PHB::GFP in the stele and its initials

in ago10-1, consistent with a protective role for AGO10 as predicted by our model. Although we could not engineer pPHB::PHB::GFP into the AGO10^{OX} background (because it already contains a GFP tag for AGO10), the time course analysis (Figure 6) was particularly insightful. It helps refine the model substantially as it reveals a striking coordination of the pincer formation, on the one hand, and the progressive concentration of the pPHB::PHB::GFP signal along the xylem axis. We made absolutely sure that identical laser settings were used all throughout the new time course analysis presented in Figure 6A, C-D and Figures EV5A, C-D, EV6). We interpret the new findings as follows (also explained in the revised main text):

At early stages (3-4 DAG), GFP:AGO10 and endoAGO accumulate to high levels inside the stele, though less so in the pericycle; GFP:AGO10 is not yet depleted from the xylem axis. PHB is lowly transcribed at these early time points and thus protected by abundant AGO10 in most of the stele. The miR165/166 gradient is thereby radial, with maximal silencing on the outer part (e.g. pericycle, containing less AGO10) and minimal silencing inside it. The ensuing picture, both longitudinally and radially, is essentially the same as reported by Miyashima et al., 2011 (Figure 4B) using the same pPHB::PGB::GFP reporter.

At later time points (5-8 DAG), however, GFP::AGO10 and endoAGO10 levels drop substantially and GFP::AGO10 progressively acquires the pincer shape. Transcript levels of PHB::GFP (as inferred by the pPHB::H2B::GFP reporter) increase strongly over the same time window. This likely explains why the intensity (as opposed to shape, which is progressively focussed along the xylem axis) of the gradient's signal remains approximately the same all along the time course. The gain in PHB transcripts throughout the stele (and beyond) now presumably provides a means to consume the abundant mobile miR165/166 via AGO1-dependent target-directed miRNA degradation (TDMD; Shi et al. 2020; 10.1126/science.abc9359). This was indeed already anticipated by Carlsbecker et al. in 2010.

We infer that the miR165/166 pool not consumed by TDMD would then be buffered by AGO10, but only within its accumulation domain (pincer), leaving a presumably small miR165/166 fraction available for the AGO10-free xylem axis, hence possibly explaining the now linear shape of the HD-ZIP III gradient at 6 DAG and beyond. Our time course analysis shows that at about 8 DAG the signal is essentially focussed in the 2/3 MX cells inside the axis (it does not evolve anymore afterwards), as we had already reported in the first version of the study. This pattern is in line with the recent *in situ* hybridization patterns of Fan et al. 2021 (<https://doi.org/10.1073/pnas.2022547118> presumably obtained at later times post-germination than those of Miyashima et al., 2011:

Thus, the time course reveals a highly dynamic process whereby the PHB gradient is being spatially refined with the help of AGO10 over a narrow yet likely decisive time frame. AGO10 is key because in its absence: (i) the PHB::GFP signal is very low and (ii) what remains of it is more widespread throughout the stele, as opposed to focussed on the axis in WT. We are hopeful that these new analyses now satisfactory address the referee's request.

6) Figure 4H. As most of roots of sdn1sdn2 show 2px and no mx, is this phenotype consistent throughout the root or perhaps the formation of metaxylem is delayed?

As indicated previously we have now imaged the xylem in both Qc-

proximal (as before) and QC-distal zones. The anomaly of the lack of MX in the sdn1sdn2 roots seems to correct itself in the QC-distal regions in most cases, as now shown in Figure 4H and Supplementary data file 3.

Supporting literature for Referee #2

Fan P, et al. The receptor-like kinases BAM1 and BAM2 are required for root xylem patterning. Proc Natl Acad Sci U S A. 2021 Mar 23;118(12):e2022547118. doi: 10.1073/pnas.2022547118.

Referee #3:

Mirlohi et al report on the roles of AGO1 and AGO10 in confining HD-ZIP III transcription patterns in the Arabidopsis root through miR165/6. There are several unanswered questions in this area that the study approaches. While there are some interesting observations, the study is somewhat incomplete and inconclusive. There are several points that need to be extended upon or clarified, listed below:

A) Firstly, the writing style is convoluted and elaborate. Many words are used to describe relatively simple and obvious observations and their implications. ***I would encourage the authors to streamline the text. Where possible, the use of less convoluted, shorter sentences would help the reader.***

We are sorry that the style in the first version was too convoluted and have now tried our best to tackle the problem in the revised version.

B) Specific points:

1) The conclusions related to AGO10-GFP expression levels relative to WT levels are problematic without proper quantification. It ***is clear that there is a gradient of accumulation levels among the lines used, but none of these seems to have WT levels.***

*We agree with the referee that the original western blot in Figure 1B merely showed a graded GFP:AGO10 accumulation among the selected lines. Note, however, that this western was made with proteins extracted from **inflorescences** in which AGO10 is much more abundant (at least 5 times) than in roots (see estimate in the new Figure 1B). This possibly blurred the differences between the selected lines and WT.*

We thus decided to perform new western analyses in the roots of these selected lines. The results are now presented in the new Figure 1B and are in line, we believe, with the statements made in the text. In particular it appears, at least in roots (organ of relevance here), that A10^{NX} accumulates GFP:AGO10 to WT-like levels, whereas A10^{OX} and A10^{UX} are respectively well-above and below these levels. We are hopeful that the new analysis now addresses the referee's valid comment.

2) ***It is strange to conclude that AGO10 "prevails" in the RAM without actually comparing AGO10 and AGO1 levels in these cells.***

We apologize for the erroneous and misleading wording used here. We just meant to explain that the RAM's stele is the main accumulation zone of AGO10 notwithstanding AGO1 accumulation (no comparison was meant between AGO1 and AGO10). We have now rephrased as:

"Therefore, AGO10 accumulates predominantly in the stele of the root apical meristem (RAM), inside which its expression overlaps with that of its closest and ubiquitously expressed paralog, AGO1".

3) ***The conclusion that post-transcriptional control determines the proximal-to-distal accumulation gradient of AGO10 in the root stele is not trivial to make. To ensure that***

the gradients are indeed different, the authors would have to quantify levels of GFP fluorescence at multiple defined positions relative to the QC and determine gradient steepness.

An alternative interpretation of their data is that the AGO10-GFP protein is simply less stable than free GFP, as is the case for many other proteins.

The reviewer is correct and we have now measured the intensity of GFP from both constructs at different zones above the QC as depicted in Figure 1C and Supplementary data file 1. We found that both form a gradient but that the GFP:AGO10 gradient is steeper than the gradient from the transcriptional reporter. We also performed comparative transient expression experiments in *N.benthamiana* leaves looking at GFP:AGO1 versus GFP:AGO10 levels. The results show that the two GFP fusion proteins have very similar steady-state accumulation levels, if not slightly higher for GFP:AGO10 (Figure EV1D). This is in stark contrast with their accumulation within the *Arabidopsis* stele where GFP:AGO1 is homogeneously longitudinally distributed whereas GFP:AGO10 forms a steep gradient from the QC-proximal region. As now explained in the main text, this difference argues that a stele-intrinsic biological process (post-transcriptional negative regulation), as opposed to selective destabilization caused by fusing GFP, likely underpins the gradient observed therein with GFP:AGO10, but not with GFP:AGO1.

Finally, that AGO10 undergoes strong and dynamic post-transcriptional negative regulation is also evident from the germination time-course analysis presented in Figure 6. The GFP:AGO10 fusion protein indeed accumulates at extremely high levels at 3 DAG but is drastically reduced at 6 DAG in a manner not commensurate with the slight drop in transcription over the same time course. It is hard to imagine how a GFP-intrinsic destabilizing effect would account for changes observed over time.

We are hopeful that the new analyses now address the referee's helpful comment.

4) The depiction of y-axes in graphs (e.g. in Figures 2B,D,E) exaggerates differences, as the origin is not 0. I would propose to clearly indicate that the axis is interrupted, and a segment is shown.

All the graphs have been redone based on new experiments now conducted on $n=30$ individuals in each genotype. For full transparency, we have presented them with the origin set at 0 without interruption. The results remain unchanged and are statistically significant in all cases.

5) When comparing AGO1 and AGO10 in their binding (affinity) to miR165/6, the strategy used is not really giving an answer. The authors do correct for the narrow versus broad expression pattern, but this does not do justice to differences that may exist within the relevant cell type. The proper experiment would have been to express AGO1-GFP from the AGO10 promoter and compare lines directly. As presented, the data is consistent but does not prove, a higher affinity of AGO10 to miR165/6, compared to AGO1.

We agree with the referee and thank them for the suggested experiment, which assumes, nonetheless, that both AGO1 and AGO10 mRNA/proteins will accumulate comparably within the AGO10 expression domain. However, the AGO1 mRNA, but not the AGO10 mRNA, is negatively feed-back regulated by miR168 (Vaucheret et al. 2006 doi: 10.1016/j.molcel.2006.03.011) . MiR168 levels are high across many root tip layers including the stele where AGO10 is mostly expressed, as shown here using root layer-specific AGO1 IPs (Brosnan et al. 2019 EMBO J; <https://www.miroot.ethz.ch>):

This would “cap” AGO1 levels in this tissue in a manner not affecting AGO10, which is not targeted/regulated by miR168.

Additionally, the AGO10 protein/mRNA, but not the AGO1 protein/mRNA, undergoes substantial post-transcriptional down-regulation in the AGO10 expression domain within the stele (Figures 1 and 6). AGO1, by contrast, does not undergo such regulation (Figure 1 and EV5B). Thus, if we were to follow the proposed strategy, many cells would ectopically accumulate the AGO1 protein within the AGO10 expression domain, which would, therefore, not truly reflect" differences that may exist within the relevant cell types".

However, given that the referee’s point is valid and that the AGO1-vs-AGO10 competition for miR165/166 is a corner stone of both our and the accompanying study by El Arbi et al. 2023, we decided to test an additional experiment to strengthen the notion that AGO10 has more affinity than AGO1 for miR165/166. We note that the referee does not question that AGO10 is selective for miR165/166, which is established by comparisons made to miR158 (most abundant pan-layer root miRNA) and miR160 (highly stele-enriched miRNA) in AGO1 versus AGO10 immunoprecipitates. To strengthen the notion that AGO10 displays competitive and higher affinity for miR165/166 over AGO1, we exploited previous report that AGO10-bound (unlike AGO1-bound) miR165/166 specifically undergoes degradation in the SAM, a process mediated by AGO10-coupled SDN1/2 (Ramachandran & Chen, 2008; Yu et al., 2017)). This translates into more miR165/166 being immunoprecipitated in AGO1 in the ago10 mutant- as opposed to WT- background (Zhu et al., 2011; Yu et al., 2017). The new data in Figure 3B support the idea that the AGO10-bound miR165/166 fraction also selectively undergoes degradation in roots. Even though AGO10 is much less abundant in the RAM than in the aerial tissues (e.g. Figure 1B for inflorescences) and displays a much narrower expression domain therein, we attempted to conduct similar IP experiments in the roots of ago10-1 versus WT plants using an antibody against endoAGO1. The new results, now shown in amended Figure 3B, are similar to those previously used to conclude that AGO10 displays higher competitive affinity for miR165/166 than AGO1 in the SAM (Zhu et al., 2011; Yu et al., 2017).

We are hopeful that the addition of this new experiment (which is also now discussed in the amended results’ text) will convince the referee that AGO10 indeed displays both selectivity and competitive affinity for miR165/166 in the RAM. Also, note that, despite the added experiment, we have replaced the wording “conclude” by “these results are consistent with” to further accommodate the referee’s comment.

6) The experiments with P19 are consistent with the model the authors propose, but it is not entirely clear which point the authors wish to make. Movement of miR165/6 has been demonstrated previously, so mopping these miR's up in the endodermis would certainly lead to loss of their role in the stele. Maybe I am missing something important here...

We are sorry that we could not convey more clearly the rationale for these experiments. We have now strived to provide a better context by including panel A in Figure 5 and amending the corresponding text’s section. The referee is only partly right in stating that “Movement of miR165/6 has been demonstrated previously”. What was shown previously is that miR165/166 acts non cell autonomously in the root (this, incidentally, still remains merely inferred in the shoot). Yet, the precise molecular form of its movement was never determined, as is the case for many mobile miRNAs. Our and others’ work indicates that in some cases, capped and polyadenylated non-processed primary miRNA transcripts (pri-

miRNAs) move. In others, the fully processed, mature miRNA (Dicer products) moves. There is limited yet equal evidence for both types of circumstances and there are foreseeable biological implications to the movement of either type of molecule (Voinnet; 2022; 10.1038/s41580-022-00455-0).

In the study's context, we contend that formation of an adequate gradient of mobile miR165/166 (which are highly abundant species) is conditioned by their "consumption", in traversed cells, by at least two AGO proteins. AGO1 would use miR165/166 for silencing the HD-ZIP IIIs, resulting in AGO1-dependent target-directed miRNA degradation (TDMD; Shi et al. 2020; 10.1126/science.abc9359) as already anticipated by Carlsbecker et al., back in 2010. AGO10 would also consume miR165/166 by inducing its SDN1/2-mediated degradation upon binding (Figure 3B). If AGO1/10-mediated consumption is key to adequate gradient establishment (an underpinning of both submitted studies), then the mobile form of miR165/166 (mature miRNA vs long pri-miRNA) does matter because AGOs do not bind (and, hence, do not consume) pri-miRNAs. As we now outline in the revised text and in the new panel of Figure 5, "pri-miR165/166 movement would bypass consumption altogether across an unspecified number of traversed cells (Figure 5A), complicating the intertwined notions of competitive binding and adequate gradient formation". The experiments conducted with P19 are a classic in the mobile sRNA field as they use the protein's highly selective and strong affinity for mature small RNA species only (Skopelitis et al. 2018; 10.1038/s41467-018-05571-0; Brosnan et al. 2019; doi: 10.15252/embj.2018100754; Brioudes et al. 2022; doi: 10.1111/nph.18180). Although the results are not black-and-white, they support movement of miR165/166 as a mature miRNA species. We hope that the added text and panel 5A now help to better understand the P19 experiments and their motivation.

7) The observations on AGO10 pattern evolution over time in the stele are very interesting, and support the notion that AGO10 patterns contribute to the observed HD-ZIP III patterns. A direct test of this role of AGO10 patterns would be to express an additional copy of AGO10 from a xylem-specific promoter. According to the model, this should directly impact HD-ZIP II patterns in a predictable manner.

We thank the reviewer for their supporting assessment of our original time course analysis. The two other referees asked us to investigate the fate of vascular HD-ZIP IIIs in the context of ago10-1 in order to bolster our model. We have now focused on pPHB::PHB:GFP for which we have built multiple lines in the WT and ago10-1 background. Their study in parallel to the pPHB transcriptional reporter, A10^{NX} and other reporters for AGO1 and miR165/166 was particularly unravelling. We believe the results now strengthen, but also refine, the original model of how AGO10 might help establish and focus the miR165/166 gradient along the xylem axis. This whole new analysis is now disclosed in Figure 6C, and discussed in the text. We hope the referee will concur that the results make a supportive case for our model.

On the other hand, the suggested experiment is very interesting in principle, but it is unclear to us how AGO10 ectopically expressed from a xylem promoter would be prevented from undergoing the same negative post-transcriptional regulation as the endogenous AGO10 or AGO10^{NX} as observed in the study. Granted, there is a slight transcriptional down-regulation, but it is not commensurate with the very substantial negative post-transcriptional control that modulates both the steady-state levels and spatial distribution of the AGO10 protein, most prominently along the axis. For the suggested experiment to be fully informative, an AGO10 allele that resists this negative post-transcriptional regulation in the axis would be needed. Since we do not know yet the mechanism involved (it is a future prospect discussed in the "concluding remarks" section), such an allele cannot be constructed, unfortunately. Nonetheless, we are hopeful that the new data highlighted here will now satisfy the referee.

Supporting literature for Referee #3

Brioudes F, Jay F, Voinnet O. Suppression of both intra- and intercellular RNA silencing by the tombusviral P19 protein requires its small RNA binding property. *New Phytol.* 2022 Aug;235(3):824-829. doi: 10.1111/nph.18180.

Brosnan CA, et al. Genome-scale, single-cell-type resolution of microRNA activities within a whole plant organ. *EMBO J.* 2019 Jul 1;38(13):e100754. doi: 10.15252/embj.2018100754.

Shi CY, et al. The ZSWIM8 ubiquitin ligase mediates target-directed microRNA degradation. *Science.* 2020 Dec 18;370(6523):eabc9359. doi: 10.1126/science.abc9359.

Skopelitis DS, et al. Gating of miRNA movement at defined cell-cell interfaces governs their impact as positional signals. *Nat Commun.* 2018 Aug 6;9(1):3107. doi: 10.1038/s41467-018-05571-0.

Vaucheret H, Mallory AC, Bartel DP. AGO1 homeostasis entails coexpression of MIR168 and AGO1 and preferential stabilization of miR168 by AGO1. *Mol Cell.* 2006 Apr 7;22(1):129-36. doi: 10.1016/j.molcel.2006.03.011.

Voinnet O. Revisiting small RNA movement in plants. *Nat Rev Mol Cell Biol.* 2022 Mar;23(3):163-164. doi: 10.1038/s41580-022-00455-0.

____*end of rebuttal*____

Dear Olivier,

We have now received re-review reports from two referees, which I have added below. As you will see, you have addressed their concerns satisfactorily. Before I can finally accept the manuscript though, there are some remaining editorial points which need to be addressed. In this regard would you please:

- acknowledge funding from the Swiss National Foundation (SNF) grants 310030 197832 in our online submission system,
- rename the conflict of interest statement as the "Disclosure and competing interests statement",
- remove the AC/CrediT section from the text,
- upload figures as individual, high-resolution files,
- uploaded datasets as 3 Excel files with multiple sheets, instead of separate Excel files zipped together,
- remove legends from the manuscript file, and upload them as separate tabs in each Excel file using the nomenclature "Dataset EV1-EV3" with the appropriate callouts,
- provide source data for figures 1A, 1D, 2A, 2C, 2E, 2H, 5A, 5B, 5D, 6A-E; label source data for figure 4 correctly; reorganize source data files to one file/folder per figure, ZIPped for each main figure. For EV and/or appendix figures,
- ZIP together all source data and complete the uploaded blank source data checklist,
- for transparency, the re-use of the same root tips between Figure 1A and Figure EV1 A and C should be detailed in the corresponding figure legends,
- summarise data description summaries (i.e. for 1a, d; 2a-b, d-e, h; 4a-h; 5b, e-f; EV 1a, c; EV 5a-b, d) at the end of legends as 'Data Information',
- correct the statistical information for figure 2f, which is incorrectly labelled as 2e, and the scale bar in the legend for figure EV 5b, which is incorrectly labelled as EV 5c,
- correct the mismatch between the annotated p values in the figure legend and the annotated p values in the figure file of figures 2b, d and f,
- define error bars in the legends of figures 4c, e, g-h; 5c and EV 3c-d,
- define scale bars in the legends of figures EV 2b; EV 3a-b, d-e,
- define empty and filled arrowheads in the legend of figure EV 3d-e,
- our production can only accept five EV figures, please move one either to the main text, or an appendix file, and
- correct the section order so that main and EV figure legends are placed after the references.

I look forward to receiving your revised manuscript.

EMBO Press is an editorially independent publishing platform for the development of EMBO scientific publications.

Best wishes,

William

William Teale, PhD
Editor
The EMBO Journal
w.teale@embojournal.org

We realize that it is difficult to revise to a specific deadline. In the interest of protecting the conceptual advance provided by the work, we recommend a revision within 3 months (12th May 2024). Please discuss the revision progress ahead of this time with the editor if you require more time to complete the revisions. Use the link below to submit your revision:

Referee #2:

The revision undertaken by the authors resulted in a substantial improvement to the manuscript. I have no additional comments at this point.

Referee #3:

The authors have addressed the points I raised in a professional manner. I support publication.

All editorial and formatting issues were resolved by the authors.

Dear Olivier,

I am pleased to inform you that your manuscript has been accepted for publication in the EMBO Journal.

Congratulations on a really elegant study!

Best wishes,

William

William Teale, PhD
Editor
The EMBO Journal
w.teale@embojournal.org
